# Major Foodborne Bacterial Pathogens in Poultry: Implications for Human Health and the Poultry Industry and Probiotic Mitigation Strategies

**DOI:** 10.3390/microorganisms13102363

**Published:** 2025-10-14

**Authors:** Shreeya Sharma, Sukhman Kaur, Mostafa Naguib, Ari Bragg, Abigail Schneider, Raveendra R. Kulkarni, Ali Nazmi, Khaled Abdelaziz

**Affiliations:** 1Department of Animal and Veterinary Sciences, Clemson University, Clemson, SC 29634, USA; shreeys@g.clemson.edu (S.S.); sukhman@g.clemson.edu (S.K.); mnaguib@g.clemson.edu (M.N.); ajbragg@g.clemson.edu (A.B.); abbyrschneider@yahoo.com (A.S.); 2Department of Poultry Diseases, Faculty of Veterinary Medicine, Cairo University, Cairo 12211, Egypt; 3Population Health and Pathobiology Department, College of Veterinary Medicine, North Carolina State University, Raleigh, NC 27607, USA; ravi_kulkarni@ncsu.edu; 4Department of Animal Sciences, The Ohio State University, Columbus, OH 43013, USA; nazmi.1@osu.edu; 5Food for Health Discovery Theme, The Ohio State University, Columbus, OH 43013, USA; 6School of Health Research (CUSHR), Clemson University, Clemson, SC 29634, USA

**Keywords:** poultry, chickens, *Salmonella*, *Campylobacter*, *Listeria monocytogenes*, *Clostridium perfringens*, *Escherichia coli*, foodborne pathogens, food poisoning, bacteria, probiotics

## Abstract

Poultry production has become the fastest-growing sector in global meat supply. However, the intensification of poultry farming has increased the risk of zoonotic transmission of bacterial pathogens such as *Salmonella* spp., *Campylobacter* spp., *Escherichia coli*, *Clostridium perfringens*, and *Listeria monocytogenes*. These bacterial agents pose major public health concerns, contributing to millions of human infections annually and substantial economic losses. Historically, antibiotic growth promoters (AGPs) were widely used to mitigate disease burden and improve poultry productivity. Yet, the global shift away from AGPs due to concerns over antimicrobial resistance has spurred interest in antimicrobial alternatives. Among these, probiotics have been explored as a promising preharvest intervention. This review investigates major bacterial foodborne pathogens associated with poultry and evaluates the practical implementation of probiotic-based strategies in modern poultry production systems, with the goal of reducing pathogen load and enhancing overall food safety.

## 1. Introduction

Poultry meat plays a vital role in the global food supply as a major source of animal protein. Its production has expanded dramatically in recent decades, from about 15.1 million tonnes in 1970 to roughly 103 million tonnes in 2024/25, making it the fastest-growing meat sector worldwide [1]. In 2024, the US produced 9.33 billion broiler chickens, yielding 61.1 billion pounds of live weight with a total production value of around $70.2 billion [2]. This rapid growth underlines poultry’s economic and nutritional importance. At the same time, intensive poultry production also heightens the risk of zoonotic transmission, as poultry flocks and products can harbor zoonotic pathogens [3,4].

Several bacterial agents, such as *Campylobacter* spp., *Salmonella* spp., *Escherichia coli*, *Clostridium perfringens*, and *Listeria monocytogenes*, are frequently associated with poultry and pose food safety risks. Bacterial-induced foodborne illnesses have a significant health and economic burden. According to the U.S. Department of Agriculture (USDA) Economic Research Service (ERS), 15 major pathogens are responsible for 95% of reported foodborne illnesses, resulting in a total economic burden of $15.6 billion. *Campylobacter jejuni* is cited as the most common bacterial cause of human gastroenteritis globally [5], where poultry products contribute to a staggering 70% of its transmission [6]. Similarly, *Salmonella* is responsible for ~94 million human gastroenteritis cases and ~155,000 deaths each year [7], making it one of the most important foodborne pathogens in poultry. Non-typhoidal *Salmonella* serotypes (e.g., *S.* Enteritidis, *S.* Typhimurium) persist widely in poultry and eggs. Contaminated table eggs are a leading source of *S.* Enteritidis, with 70% of human infections attributed to this serovar [8]. *L. monocytogenes* is less common in poultry, but it causes invasive listeriosis with roughly a 20% case-fatality rate. Sporadic listeriosis cases have been traced to contaminated poultry, being isolated at multiple points in the poultry production environment [9]. Virulent strains of *C. perfringens* cause necrotic enteritis (NE) in chickens, an intestinal disease that can cause poultry mortality up to ~30%, costing the global poultry industry over $6 billion annually [10]. While *C. perfringens* strains in poultry differ from classic food poisoning strains, the heat-resistant spores can survive cooking and increase carcass contamination. Finally, pathogenic *E. coli* strains, particularly avian-pathogenic *E. coli* (APEC), cause colibacillosis in birds, inflicting heavy production losses. In the US alone, APEC infections are estimated to cause roughly $40 million per year in losses from condemned birds [11]. From a food-safety standpoint, improper handling or undercooking allows these strains to cause urinary tract infections in humans.

The combined impacts of these pathogens create a dual burden. For poultry producers, disease outbreaks lead to direct losses: high mortality, reduced growth or egg production, and costs for treatment, culling, or sanitation. For example, *E. coli* and *Salmonella* outbreaks often force whole flock culling or product recalls. One recent report in 2023 estimates that in the US, foodborne illnesses alone cost about $75 billion annually in economic loss, in large part due to bacterial pathogens. Among these major pathogens, *Salmonella* alone accounts for $17 billion, *Campylobacter* for $11 billion, *Listeria* for $4 billion, diarrheagenic *E. coli* for $3.7 million and *C. perfringens* for $343 million in associated costs [12,13]. The impact of these infections is widespread, with an estimated 49 million cases occurring annually in the US, affecting approximately 15% of the population. Reports indicate that foodborne pathogens caused approximately 350 million cases of illness globally in 2010, with food animals serving as asymptomatic carriers of these bacteria. Among the 86 known transmissible diseases from animals to humans, 20 are bacterial in origin [14]. Figure 1 illustrates the different pathways of transmission for these foodborne pathogens.

Historically, antibiotic growth promoters (AGPs) were used to mitigate disease burden, owing to their benefits in reducing subclinical infections and reducing mortality rates, as well as enhancing feed efficiency and growth, in production facilities [15]. Although antibiotics serve important therapeutic purposes, their prolonged sub-therapeutic use (low doses administered over extended periods) is associated with public health concerns, such as antibiotic resistance, environmental antibiotic pollution and antibiotic residues in animal-derived products [16]. The US, EU, Canada, and Mexico, among several leading countries, have placed bans or strict limitations on subtherapeutic use of antibiotics as growth promoters [17].

Thus, alternative preharvest interventions are urgently sought [3,18,19,20,21,22,23]. Among these, probiotic feed additives have attracted attention. Probiotics are defined as “live, beneficial microbes that confer health benefits to the host.” [24]. Probiotics work by multiple complementary mechanisms to inhibit pathogens in the gut and improve gut health. Probiotic bacteria colonize the gut and occupy niches or adhesion sites, outcompeting pathogens for nutrients and space [18,19,20,25,26]. Many probiotics (especially lactic acid bacteria) secrete organic acids (lactic, acetic, etc.) and bacteriocins that directly inhibit pathogens [24,27]. Additionally, they produce useful metabolites, enzymes and induce mucin and defensin secretion, improve intestinal integrity, helping prime both innate and adaptive immunity against enteric pathogens [28,29].

Probiotic species belonging to *Lactobacillus*, *Streptococcus*, *Bacillus*, *Bifidobacterium*, *Enterococcus*, *Aspergillus*, *Candida*, and *Saccharomyces* have been proven to exert beneficial effects on broiler production [29]. In the US, probiotics are generally marketed as GRAS (Generally Recognized As Safe) feed additives or animal health products [30]. The selection of probiotics for use in poultry relies on a set of criteria to ensure both biological efficacy and industrial feasibility. Probiotic strains are often inhabitants of the gut microbiota, but they cannot be called ‘probiotics’ until they are well-characterized and demonstrate health benefits both in vivo and in vitro [24]. Selection criteria involve rigorous testing, such as evaluation of acid and bile salt tolerance, antimicrobial activity, adhesion capacity to intestinal epithelial cells, and viability during storage and processing [27,31,32,33,34,35]. Moreover, practical factors like appropriate dosing (≥10^6^–10^9^ CFU per bird), production stage of the bird and compatibility with other feed additives such as prebiotics should be taken into consideration. In this review, we highlight the significant burden posed by major bacterial foodborne pathogens in poultry, namely *Salmonella*, *Campylobacter*, *E. coli*, *L. monocytogenes*, and *C. perfringens* and underscore probiotic control strategies to enhance poultry health and food safety.

## 2. Major Food-Borne Pathogens in Poultry and Probiotic Mitigation Strategies

### 2.1. Salmonella

Salmonellosis is a foodborne disease that primarily causes diarrhea in humans [5]. Globally, non-typhoidal salmonellosis (NTS) has been reported to cause about 93 million cases of gastroenteritis and 155,000 deaths annually [34]. The Centers for Disease Control and Prevention (CDC) has estimated that it is responsible for 1.35 million infections and 420 deaths annually in the United States [35]. Estimates from the WHO Foodborne Disease Burden Epidemiology Reference Group (FERG, 2007–2015) show that NTS and invasive non-typhoidal salmonellosis (iNTS) are responsible for 4.38 million and 3.9 million Disability-adjusted life years (DALYs), respectively [36], a metric commonly used to measure the impact of a disease or illness on population health. Recently, the Global Burden of Disease (GBD) reported that iNTS caused 594,000 infections, leading to 79,000 deaths and 6.11 million DALYs worldwide [37]. This overlays an annual economic burden of $3.31 billion, with disease severity in humans varying by strain, health status and age.

Poultry meat and eggs are the primary carriers of *Salmonella* [5], and they cause an annual loss of $64 to $114 million to the US poultry industry. Between 1998 and 2008, poultry was linked to 17.9% of foodborne illnesses in the U.S., with *S.* Enteritidis and *S.* Typhimurium responsible for 17% and 34% of poultry-related infections, respectively [38].

#### 2.1.1. *Salmonella* Virulence, Pathogenicity and Mode of Transmission

*Salmonella* is a gram-negative, oxidase-negative, non-spore-forming, facultative anaerobic bacterium belonging to the Enterobacteriaceae family [35]. Its cell wall is composed of lipids, lipopolysaccharides, proteins and lipoproteins [39]. The bacterial surface possesses a polysaccharide somatic (O) antigen, composed of short oligosaccharides [40]. *Salmonella* is classified based on roughly 60 somatic antigens. Additionally, it possesses proteinaceous and heat-labile flagellar (H) antigens. Based on agglutination reactions of O and H antigens, approximately 2643 serotypes of the pathogen were recognized by 2000, a number that has risen to nearly 3000 by 2017 [41,42].

The genus *Salmonella* is divided into two species, *S. enterica* and *S. bongori* [41,43]. Among them, *S. enterica* is of primary clinical and veterinary importance, and is divided into six subspecies: *S. enterica* subsp. *enterica* (I), *S. enterica* subsp. *salamae* (II), *S. enterica* subsp. *arizonae* (IIIa), *S. enterica* subsp. *diarizonae* (IIIb), *S. enterica* subsp. *houtenae* (IV), and *S. enterica* subsp. *indica* [41,44]. These species are further divided into serovars, which differ in host adaptation, pathogenicity and epidemiological distribution. *Salmonella* serovars are classified as typhoidal and non-typhoidal based on the type of disease they cause. Typhoidal serovars such as *S.* Typhi and *S.* Paratyphi, are strictly adapted to human hosts and are responsible for causing enteric fever, commonly known as typhoid fever, through the fecal-oral route. Contrastingly, non-typhoidal *Salmonella* is zoonotic and can infect many animals, including birds, reptiles, dogs, cats, rodents, sheep, poultry, and pigs [45]. The common non-typhoidal serovars reported in human and poultry infections include *S.* Enteritidis, *S.* Typhimurium, *S*. Heidelberg, *S.* Kentucky, and *S.* Gallinarum [46]. Among these, *S.* Enteritidis is the most common serovar linked to human salmonellosis outbreaks and is strongly associated with vertical transmission in hens, contaminating eggs and egg products [47]. 70% of *S.* Enteritidis infections are caused by grade A table eggs. *S.* Typhimurium has a broader host range, infecting many warm-blooded animals besides poultry, and is commonly found in poultry meat and slaughter environments [8]. Additionally, *S.* Heidelberg is frequently isolated from all poultry production systems in North America and increasingly associated with multidrug resistance and human salmonellosis. *S.* Kentucky is present in dairy cattle and broiler chickens [48]. *S.* Gallinarum is host-restricted to birds, causing fowl typhoid and pullorum disease, which resulted in severe losses to the poultry industry in the US; however, it was entirely eradicated by the 1960s due to the continual efforts of the National Poultry Improvement Plan (NPIP) [47,49].

Contaminated broiler chicken meat is a major source of *Salmonella* infection in humans. Hassanein et al. (2011) tested 75 samples of beef and chicken and observed that 20% of minced frozen beef, 36% of frozen chicken leg and 52% of the frozen chicken fillets were contaminated with prominent *Salmonella* serovars like *S.* Enteritidis and *S.* Kentucky [50]. Table 1 summarizes the classification of *Salmonella* serovars.

*Salmonella* can be transmitted to poultry via two routes: horizontal and vertical transmission, with horizontal being the most common. In the horizontal transmission, *Salmonella* spreads between birds through the fecal-oral route, skin wounds, blood, mating, and contaminated equipment [55]. Vectors, including cockroaches, poultry mites (*Dermanyssus gallinae*), litter beetle (*Alphitobius diaperinus*), lizards and rodents, are sources of *Salmonella* infection in poultry [56,57,58]. Several wild birds like passerine birds also serve as asymptomatic carriers of *Salmonella*, transmitting *Salmonella* serotypes to poultry during migration, seasonal movements and feeding [59,60,61]. On the other hand, vertical or transovarial transmission occurs when bacteria from infected hens are transferred to their hatched chicks. *Salmonella* invades reproductive organs, reaching the yolk, vitelline membrane and albumen before eggshell formation and laying [62]. The digestive tract of chickens is recognized as a primary multiplication site for *Salmonella*, from where the organism is excreted through feces, leading to widespread environmental contamination [63].

These infected chickens and their products, including contaminated meat and eggs, are the primary source of *Salmonella* infection in humans. Factors like serotype, infective dose of the pathogen and immune status of the host determine the severity of disease in humans. Usually, children below five years of age and immunosuppressed individuals are more vulnerable to infection [64]. Upon infection with pathogenic strains of *Salmonella*, the host phagocytic cells, mainly macrophages, engulf the bacterial cells, where they get encapsulated in the vacuole. While recognizing bacterial components by host cells typically triggers lysosomal fusion and enzymatic degradation to eliminate invading microbes, *Salmonella* exhibits resistance to this process by injecting effector proteins into the vacuole through the Type III secretion system, consequently altering the compartment structure. This altered vacuole structure prevents lysosomal fusion and allows the bacteria to survive and multiply within phagocytic cells, causing gastroenteritis [65]. NTS infections can also cause cholecystitis, pancreatitis and appendicitis [39]. Some serotypes like *S.* Dublin and *S.* Cholearsuis can enter the bloodstream and cause bacteremia, a condition associated with high fever and, in severe cases, can lead to septic shock and other complications, such as urinary tract infections, pneumonia, endocarditis and meningitis [7,66]. Classification of *Salmonella* pathogenicity islands (*SPIs*), the regions on the *Salmonella* plasmid encoding several virulence factors, and the virulence factors are given in Table 2 and Table 3, respectively.

#### 2.1.2. Probiotic Efficacy Against Salmonellosis in Poultry

Several preharvest and post-harvest interventions have been adopted to combat *Salmonella* infections in poultry. Preharvest interventions include management at the farm level, use of feed additives and biosecurity measures. Biosecurity measures include site cleanliness, immunization, boot sanitization, hand hygiene, better rodent, fly and red mite control, and disinfection between the flocks. Antibiotic incorporation is another pre-harvest strategy that has been used in poultry since the 1940s due to its rapid and effective action in controlling enteric pathogens and enhancing the growth performance of the host [81]. Although antibiotics like penicillin, tetracycline, enrofloxacin, ciprofloxacin, azithromycin, ceftriaxone, chloramphenicol, neomycin, polymyxin, nitrofurazolidone, amoxicillin, and chloramphenicol have been used to treat salmonellosis [35,82,83], non-antibiotic-based approaches like probiotics, prebiotics, synbiotics, postbiotics and phytobiotics offer low toxicity, high safety, efficient metabolism, and no environmental contamination. The bioactive derivatives of these compounds, such as unsaturated fatty acids, proteins, polysaccharides, and alkaloids, can be metabolized by the host and its gut microbiota, thereby reducing overall metabolic load on the liver and kidneys. These natural antimicrobial alternatives also disturb bacterial sensing mechanisms by altering biofilm permeability, which affects bacterial survival and growth in host cells. Additionally, they promote the secretion of antibacterial compounds such as inhibitory peptides, defensins, lysozyme and colloidal mucin by the host immune cells, as well as activating the absorption of nutrients in the intestine and balancing the synthesis of intestinal hormones [84,85,86]. Owing to these benefits, there is an increasing interest in the utilization of feed-based additives like probiotics against *Salmonella* infections in poultry [87,88,89].

Probiotic bacteria belonging to the genera *Bifidobacterium*, *Lactobacillus*, *Bacillus*, *Pediococcus* and *Enterococcus* are used to control salmonellosis in poultry [89]. Their demonstrated efficiency lies in their ability to restore gut microbiota and increase the accumulation of short-chain fatty acids (acetate, butyrate and propionate) in *Salmonella*-infected chickens. Moreover, probiotics can improve the effectiveness of *Salmonella* vaccines used in poultry, which intensifies the defensive effects of the body’s immune system [90].

Over recent years, *Bacillus* probiotics have been recognized as effective control agents for salmonellosis in poultry [28,91,92,93]. Spore formers are highly resistant to harsh and dry physical and chemical conditions and bile concentrations and can withstand the intestinal temperatures of the chickens [94]. *B. subtilis* is one of the most used *Bacillus* species in the animal feed industry [95]. It is an effective growth promoter and modulator for the gut microbiota population and the innate immune response of broiler chickens [96,97]. *B. subtilis* produces certain antimicrobials and favors the growth of gut-beneficial microorganisms that reduce invasive pathogens by CE [96]. A previous study showed that the dietary supplementation of *B. subtilis* QST-713 in *Salmonella* Gallinarum-challenged chickens improves their growth performance by modulating gut mucosal microarchitecture, including villus height. The treated birds also had less disease severity, a lower mortality rate and improved weight gain [98]. In a study conducted by Knap et al. (2011), the inclusion of *B. subtilis* (DSM17299) resulted in a 3-log reduction in cecal *S.* Heidelberg counts in birds, with a transient decrease in fecal shedding of the pathogen [99]. Moreover, *B. subtilis* can improve systemic immune responses by producing α and β-defensins, supporting host innate immunity [100], promoting immunoglobulin synthesis and improving chicken intestinal microarchitecture. Likewise, Sikandar and colleagues observed an increased bursa follicular area (165.757 μm^2^), delayed B-cell apoptosis, increased thickness of the thymus cortex and medulla and higher antibody titers against Newcastle disease virus (NDV) in *S.* Gallinarum-infected chickens receiving *B. subtilis*-inoculated diet [101]. Contrastingly, birds fed a regular diet had a bursa follicular area of about 59.124 μm^2^, decreased thymus medulla thickness and reduced germinal area of the spleen. *B. methylotrophicus*, another novel *Bacillus* species, possesses probiotic characteristics; therefore, it is used as a feed supplement in poultry as well as livestock production. For instance, in a study, the feed supplementation of either *B. subtilis* RX7 or *B. methylotrophicus* C14 at 0.1% of the diet (~1 × 10^9^ CFU/g) resulted in a significant reduction of *Salmonella* counts in the excreta in *S*. Gallinarum-infected (1 mL containing 10^8^ CFU/mL) Hy-Line Brown laying hens [102]. The authors also reported an increase in the beneficial intestinal microbiota, such as *Lactobacillus*, of the treated hens. Hosseindoust and colleagues also observed an improved eggshell strength and thickness, increased intestinal populations of *Lactobacillus* spp., and a reduction in cecal and fecal shedding in *S*. Gallinarum-infected Hy-line laying hens receiving a *B. subtilis* (strain B2A) supplemented diet [103].

*Bacillus licheniformis* can inhibit enteric pathogens in poultry by secreting antimicrobial surfactant molecules, serine protease, and other enzymes [104]. As a probiotic, *B. licheniformis* has been shown to improve bird performance, stimulate immunoglobulin production and enhance mucosal barrier function [27], and it can effectively interfere with colonization by pathogenic microbes, including *Salmonella* species. A recent study found that dietary inclusion of *B. licheniformis* and *B. subtilis* to broilers challenged with *S.* Enteritidis resulted in 0.73, 1.59 and 1.32 log decreases in cecal *Salmonella* CFU/g at 5, 12 and 21 days, respectively, post-infection [105]. Similar results were obtained in a later study conducted by [106], where feeding the chickens infected with *S.* Enteritidis a diet supplemented with a mixture of three strains of bacterial probiotic led to a 1.08 log_10_ reduction in cecal *Salmonella* content as compared to the control.

*Bacillus coagulans* is also commonly used as a probiotic, producing bacteriocin-like substances (coagulin) against various pathogens, including *Salmonella*. It can also enhance intestinal barrier integrity and gut morphology [107,108]. Zhen et al. (2018) studied the effect of *B. coagulans* in broiler chickens challenged with *S.* Enteritidis [109]. They observed that its supplementation increased alkaline phosphatase activity, villous height and the number of goblet cells in the jejunum [109]. Moreover, the birds fed a supplemented diet had 0.24, 0.41, and 0.24 log10 less *Salmonella* counts after 7, 17 and 31 days of infection compared to birds receiving a regular diet.

*Bacillus cereus* has also been observed to stimulate immune responses, typically macrophages, by enhancing phagocytosis, nitric oxide production and secretion of pro-inflammatory cytokines [110]. It also improves the digestion and absorption of food, which eventually results in improved FCR and broiler weight, as studied by Vilà et al. (2009) [111]. The results of this study indicated that the addition of Toyocerin (containing 10^9^ viable spores/g of *B. cereus* var. *toyoi* NCIMB 40112/CNCM I-1012 per gram) in the feed decreased cecal *Salmonella* load by 100% (42 days post-infection) and improved average daily weight gain (ADG) by 3.4 g, broiler weight (BW) by 141 g and feed conversion ratio (FCR) by 0.060kg/kg in *S.* Enteritidis-infected broilers.

*Bacillus amyloliquefaciens* also possesses antimicrobial activity as it can secrete several enzymes, such as barnase, cellulase, protease, amylase and xylanase [112]. For example, Poudel et al. (2025) observed a respective 0.40 and 0.60 log_10_ reduction in cecal *Salmonella* content (after 7- and 17-days post-infection) when the birds inoculated with *S.* Enteritidis were fed a diet containing a mixture of three strains of dried *B. subtilis* and two strains *B. amyloliquefaciens* in different concentrations (1 × 10^6^ CFU/kg diet (PRO1) and 2 × 10^6^ CFU/kg diet (PRO2) [113].

*Pediococcus pentosaceus* GT001 is another promising probiotic strain. When it was included in the diet of *S.* Typhimurium-infected chickens, it decreased mortality by 6.7% in treated chickens compared to the birds fed with a regular diet, where this decrease was only 3.3%. Beyond this, it also improved growth performance, immune function and intestinal morphology in broilers and brought a 1.2 log reduction in *Salmonella* count at 14 days post-infection as compared to untreated birds [114]. Khochamit et al. (2020) also conducted a study to examine the synergistic effect of bacteriocin-producing strains of *B. subtilis* KKU213 and *P. pentosaceus* NP6 in improving the growth, microbial community and health in broilers [115]. The authors observed that feeding a probiotic-inoculated diet to broilers decreased mortality; however, no *Salmonella* was detected in the treated birds (on day 18 post-treatment) compared to the control group, where *Salmonella* was 20%.

Dairy propionibacteria are a group of bacterial species involved in dairy technology and offer in vivo benefits [116]. Their probiotic properties lie in the fact that they can survive and maintain metabolic activity in the gastrointestinal tract. They produce bifidogenic molecules such as 1,4-dihydroxy-2-naphtoic acid (DHNA) and 2-amino-3-carboxy-1,4-naphthoquinone (ACNQ) that promote the growth of intestinal bifidobacteria and inhibit the growth of toxin-producing *Bacteroides* spp. and *Clostridium difficile* [117]. It was reported that *P. freudenreichii*, when administered to *S.* Heidelberg GT2011-infected turkeys, reduced the pathogen levels in cecal content by about 1 to 2 log CFU/g. Although the effect lasted only two days, supplementation with probiotics increased lactobacilli and Ruminococcaceae after 2 days. It increased the genera *Lactococcus*, *Erysipelatoclostridium*, *Leuconostoc*, and *Butyricicoccus* after 7 days [118]. It was later reported that in treatments of turkeys with *P. freudenreichii* B3523, the dissemination of *S.* Heidelberg in the liver and spleen was reduced from 20% to 60% depending on the age of animals and treatment time [118]. A recent study also observed that *P. freudenreichii* B3523 significantly reduced *Salmonella* dissemination in the liver and spleen of turkeys challenged with drug-resistant field turkey isolates of *S.* Agona, *S.* Saintpaul, and *S.* Reading. When combined with a *Salmonella* vaccine, it eliminated the pathogen from these organs. This synergistic effect of the vaccine and probiotic was found to be more effective than the vaccine alone or when used in combination with *Ligilactobacillus salivarius* in combating multidrug-resistant *Salmonella* strains in turkeys [119].

Among Lactic Acid Bacteria (LAB), *L. plantarum* has the largest genome and is commonly used in fermented foods [120]. A previous study observed that administering a mixture of *L. plantarum* and *B. subtilis* decreased the mortality rate of *S.* Typhimurium-infected chickens and increased body weight gain and feed conversion ratio. When used individually or in combination, it resulted in the complete elimination of *Salmonella* in the liver, spleen and heart of the chickens at 28 days post-infection [121]. Similarly, other LAB strains have also been used as probiotics in poultry feed. In a study conducted by Smialek et al. (2018), the incorporation of probiotic Lavipan (consisting of four strains of LAB, i.e., *Lactococcus lactis* IBB 500, *Carnobacterium divergens* S-1, *L. casei* ŁOCK 0915, *L. plantarum* ŁOCK 0862 and a yeast *Saccharomyces cerevisiae* LOCK 0141 resulted in a 98.84% reduction in cecal *Salmonella* content at 42 days of age as compared to 58.29% reduction by untreated birds [122]. In a later study, Adhikari et al. (2019) evaluated the impact of in-feed supplementation of *L. plantarum* (0.05% and 0.10% (*w*/*w*) on fecal shedding, cecal and internal organ colonization of *S*. Enteritidis-infected (2.8 × 10^8^ CFU/mL) White Leghorn laying hens [123]. The authors reported no significant reduction in cecal *S*. Enteritidis colonization 7 dpi since the mean counts in all challenged groups were around 3.7 log_10_ CFU/g, but a downregulation of IFN-γ and upregulation of IL-6 and IL-10 were observed in the group receiving a higher dose of probiotic (0.10% (*w*/*w*)).

*L. acidophilus* is another widely used LAB that can tolerate low intestinal pH, competitively excludes enteric pathogens by competing with them for nutrients and attachment sites and modulates immune mechanisms. Similarly, *L. salivarius* has been known for its immune-modulatory and anti-inflammatory properties when used as a probiotic. Bielecka et al. (2010) compared the effects of the incorporation of different strains of LAB, including *L. acidophilus* strain BS, *L. salivarius* strain AWH, *L. helveticus* strain b9, *Bifidobacterium longum* strain KNA1 and *Bacillus longum* in *S.* Enteritidis-infected chickens [124]. The authors found that these probiotic strains promoted superoxide anion production, stimulated leukocyte multiplication, and increased lysozyme and γ-globulin levels in chickens.

In an earlier study, Khaksefidi and Rahimi (2005) also studied the impact of a multi-strain probiotic diet supplementation consisting of *L. acidophilus*, *L. casei*, *Bifidobacterium bifidum*, *Aspergillus oryzae*, *Streptococcus faecium* and *Torulopsis* spp. on chicken health. They observed that feeding chickens with a probiotic-inoculated diet increased their weight and decreased the mortality rate by 3.2% compared to the control group [125]. Moreover, only 40% of carcass meat from treated chickens tested positive for *Salmonella* compared to 100% in the untreated group.

Furthermore, yeasts are another probiotic candidate known for their antimicrobial properties. Among all species, *S. cerevisiae* is one of the most frequently used strains to inhibit enteric pathogens. In an earlier study conducted by Mountzouris et al. (2015), it was observed that the *Salmonella* load on breast skin and cloacae of *Salmonella* enterica birds fed *S. cerevisiae* var. *boulardii* CNCM I-1079 was approximately 28.6% and 33.3% less, respectively, compared to birds fed a standard diet [126]. Similarly, in a later study, it was reported that supplementing probiotics containing *B. amyloliquefaciens*, *B. licheniformis*, and *B. pumilus* (454 g/ton) and yeast culture (1133 g/ton) to the feed of *S.* Enteritidis-infected Hy-Line Brown pullets resulted in 0.79 log_10_ and 0.86 log_10_ reduction in cecal *Salmonella* content, respectively, as compared to the control groups [127]. However, in a study conducted by El-Hamd and Hans (2016), the use of a probiotic blend comprising *L. plantarum*, *L. acidophilus*, and *S. cerevisiae* in the drinking water of S. Enteritidis-infected broilers didn’t reduce cecal *Salmonella* colonization, as the treated birds still harbored the pathogen [128].

Table 4 highlight the promising role of both bacterial and yeast-based probiotics in reducing *Salmonella* colonization and improving growth performance in poultry, supporting their integration as single-strain or multi-strain probiotics into pre-harvest food safety strategies.

### 2.2. Campylobacter

The risk of transmission of foodborne illnesses, such as *C. jejuni*, to humans through contaminated poultry meat and its products is considered a significant challenge for sustainable poultry production. Thermophilic *Campylobacter* species, mainly *C. jejuni* and *C. coli*, are commonly found in wild birds and domestic fowl (e.g., chickens, turkeys, ducks, and geese) [63,130]. The detection rate of *Campylobacter* might reach 100% in broiler flocks at slaughterhouses [63]. Despite the high colonization rate (up to 10^9^ CFU/g of cecal contents), *Campylobacter* infections generally cause little or no clinical disease in chickens [131]. However, the transmission of this pathogen through contaminated poultry products to humans is estimated to cause 1.5 million foodborne infections annually in the US [132]. Though *Campylobacter* is a self-limiting disease in humans, infection in immunocompromised individuals could lead to severe health complications, such as reactive arthritis, inflammatory bowel disease, and Guillain-Barré syndrome [133] with a hospitalization rate of 10% [134] and 0.2% deaths [135]. The health care costs incurred by this disease in humans were initially estimated at two billion dollars in the US [136]. However, a recent study revealed a substantially increased burden of 11 billion dollars annually [137]. Humans are primarily infected through undercooked poultry meat (50–80%), contaminated food and water or direct contact with infected animals [138].

With the lack of *Campylobacter* vaccines, other strategies, including feed additives, are being explored to control this pathogen in poultry. Due to their natural ability to inhibit pathogens and modulate host immunity, probiotics have emerged as a promising solution [139,140,141].

#### 2.2.1. *Campylobacter* Virulence, Pathogenicity and Mode of Transmission

Upon ingestion of contaminated feed and water, *Campylobacter* colonizes the avian gastrointestinal tract within 24 h, with a small infectious dose of 35 CFUs that can successfully colonize chickens’ guts [142]. Flagellum is a key structure that facilitates *C. jejuni* motility, adherence and invasion [143]. With its spiral shape, bipolar flagella enable *C. jejuni* to move in a corkscrew motion and facilitate its penetration of the mucin viscous layer [144]. Chemotaxis is a crucial process that guides *C. jejuni* to favorable sites for colonization. *C. jejuni* exhibits chemotactic responses to a variety of attractants, including intestinal mucins, amino acids, carbohydrates, and organic acid salts [145]. To identify these chemical signals, the bacterium uses methyl-accepting chemotaxis proteins (MCPs), including the colonization protein B (DocB) determinant and transducer-like protein 1 (Tlp1). The signal transduction between MCPs and the flagellar motor is mediated by the chemotaxis-regulating protein Y (CheY) [146]. A physical barrier comprising a mucus layer that lines the intestinal epithelium with secretory IgA and antimicrobial peptides, allowing commensal microbes to thrive but preventing excessive interaction [147]. The activity of flaA is downregulated with the viscosity of this layer. To overcome this, *C. jejuni* modifies its flagella and successfully penetrates the mucus layer [148]. *C. jejuni* adhesion to epithelial cells necessitates several factors, such as intact flagella, *Campylobacter* adhesion to fibronectin (CadF), and lipooligosaccharide (LOS). Following adhesion, the flagella apparatus increases the secretion of *Campylobacter* invasion antigen (Cia) [143]. Simultaneously, LOS enables *C. jejuni* to evade the host immune response in humans and facilitate its invasion [149].

Despite chickens asymptomatically harboring a high load of *C. jejuni* in their gut, upon ingesting a small dose of 500–900 CFUs in contaminated poultry products, humans experience severe diarrheal illness. The underlying mechanisms behind this controversy of the pathogenicity between human and avian hosts are not entirely understood. Initial evidence underscores mucus composition, acidity and function as a key point in these diverse outcomes [150]. Byrne et al. (2007) conducted an in vitro study that showed the ability of *C. jejuni* to adhere and invade epithelial cells [151]. The same group further explored the effect of chicken and human crude mucus on *C. jejuni* internalization. Chicken mucus was found to reduce the binding and internalization of *C. jejuni* into human epithelial cells, whereas human mucus enhanced both processes. Unique O-linked glycan structures seen in chickens’ mucins directly disrupt the adherence of *C. jejuni*. With lesser inhibitory effects shown from small intestinal and cecal mucins, purified chicken mucins from the large intestine diminish bacterial binding to human colonic epithelial cells (HCT-8) by 60–70% [152]. While the MUC2 backbones are similarly structured in human mucins [153], the absence of specific glycans leaves the mucus vulnerable to *C. jejuni* invasion [147]. Additionally, pathogen surface proteins are effectively impeded by the extensive sialylation and fucosylation patterns observed in chicken mucins [152]. It is hypothesized that the divergent body temperature of chickens (42 °C), compared with human temperature (37 °C), crucially influences the pathogenicity of *C. jejuni* in both hosts [150]. Human temperature facilitates the expression of virulence factors, such as the zinc exporter *CzcD*, to counteract the host defense mechanism [154]. However, Zhang and his team revealed that the upregulation of the *CadF* gene, which controls cell adhesion, is upregulated by *C. jejuni* at 37 °C and 42 °C, thus demonstrating that *C. jejuni* can adhere to both human and chicken intestinal cells [155]. Nevertheless, these gene expression variations do not fully explain the distinct inflammatory pathogenesis of *C. jejuni* in humans and its near-symbiotic colonization in chicken hosts. It is well-documented that host-specific immune response could justify the paradox in the *C. jejuni* virulence in humans vs. chickens. Most *C. jejuni* isolates encode for cytolethal distending toxin (CDT). However, immune cells produce neutralizing antibodies against this toxin, implicating its role in human pathogenesis [156]. Additionally, CDT-negative *C. jejuni* mutants successfully colonize the gut of chickens as wild-type strains, confirming the previous outcome [157]. At the same time, chickens may experience induced proinflammatory cytokines post-*C. jejuni* inoculation without further heterophil infiltration, robust recruitment of neutrophils and macrophages initiated by TLR-4-mediated recognition of LOS was observed in humans [158]. Another key mechanism by which *Campylobacter* evades the human immune system is the structural mimicry of *C. jejuni* LOS and nerve gangliosides, which can ultimately trigger Guillain-Barré syndrome; however, this immune evasion is not observed in chickens. Overall, in chicken infections, it seems that *C. jejuni* resides in the mucus layer without exposure to intestinal epithelial cells with a state of immunological tolerance, while in humans, it invades the intestinal layer and provokes acute disease. Collectively, it could be concluded that host-specific mucus composition, body temperature, gene expression profile, and immune responses lie behind the contrast of pathogenicity reported in human vs. chicken hosts. Virulence factors associated with *C. jejuni* pathogenicity are summarized in Table 5.

#### 2.2.2. Probiotic Efficacy Against Campylobacteriosis in Poultry

Several pre-harvest mitigation strategies against *Campylobacter* have been investigated in poultry, including biosecurity, bacteriophage therapy, vaccination, bacteriocins, prebiotics, and probiotics [139,141,167]. The limited abilities of the investigated vaccines to elicit cross-protective immunity against various strains of *Campylobacter* lie in the antigenic variability among these strains and the unsuccessful identification of the antigenic protein/s that provide complete and cross-protection against these strains. These factors remain the primary reasons why effective and reliable immune interventions are still elusive [139,168]. As such, continued investigations are still required to improve our comprehension of the complexity of *C. jejuni*-host interaction and develop more resilient and broadly applicable interventions to curb *C. jejuni* infection in chickens. There is no evidence of *Campylobacter* developing resistance to probiotics, making probiotics a viable long-term strategy for mitigating this infection. However, candidate isolates should be thoroughly screened to ensure their non-pathogenic nature and effectiveness.

Numerous laboratory tests have been developed to investigate the appropriate probiotic candidate (Table 6). The laboratory screening of probiotic isolates is crucial to assess their ability to resist the extreme acidity conditions in chickens’ gut, bile salts, and enzymatic activities and, more importantly, to detect their ability to reduce the growth, invasion, and adhesion of targeted pathogens, such as *Campylobacter*. The well diffusion assay is mainly applied to evaluate the antagonistic effect of the probiotics’ cell-free supernatants. The presence of inhibitory metabolites. such as organic acids and bacteriocins, which are associated with producing a clearance zone on the agar. On the other hand, agar spot and co-culture suspension are primarily employed to test a live probiotic culture. Despite the beneficial outcomes of testing live probiotic culture using the co-culture suspension and agar spot, it ignores the complexity of the interactions occurring within the host. Consequently, epithelial cells are a more effective screening approach to measure the probiotics’ antagonistic effect against pathogenic bacteria such as *Campylobacter*. Both candidate probiotics and pathogenic bacteria are cultured with the intestinal monolayer, and subsequently, the adhesion and invasion indices are calculated. A primary concern when using probiotics is the risk of the transfer of antibiotic-resistant genes. Hence, an antibiotic sensitivity assay is widely used to confirm the absence of antibiotic-resistance-associated genes among candidate probiotics.

Several researchers have investigated the effectiveness of probiotics against *Campylobacter* infection in chickens. However, a significant variation in the results was recorded. Several factors contribute to this observed inconsistency, including strain-specific effects, variations in bird age and type, probiotic dosage and combinations, administration routes, dosage, and duration of treatment. Moreover, management and environmental factors, such as housing type and dietary program, may influence outcomes. Understanding these variables is essential for optimizing probiotic use to control *Campylobacter* in poultry, improve food safety and reduce the risk of human transmission.

LAB group includes several genera, such as *Leuconostoc*, *Lactococcus*, *Enterococcus*, *Pediococcus*, *Streptococcus*, *Lactobacillus* and *Bifidobacterium*. While in vitro assays offer valuable insight into the probiotic potential of *Lactobacillus* species and their antagonistic effects against *Campylobacter*, their efficacy has not been replicated in vivo models. For instance, a recent study showed that although *L. sakei* L14 demonstrated strong anti-*Campylobacter* activity in vitro, it failed to reduce *C. jejuni* counts when administered orally to broiler chickens [177]. In broiler chickens, oral administration of *L. salivarius* SMXD51 on day one of age, followed by repeated doses every 2–3 days, resulted in a reduction of *C. jejuni* counts by 0.8 log_10_ and 2.8 log_10_ on day 35, respectively [178]. In contrast, dietary supplementation of the same species to mule ducks for 79 days failed to reduce *Campylobacter*’s burden [179]. When microencapsulated bacteriocins, a byproduct of *L. salivarius*, were added to the feed, they reduced *C. jejuni* counts by 4 log units after three consecutive days of administration in turkeys [180]. In another study comparing the effects of various *Lactobacillus* species, oral administration of *L. plantarum* PA18A on days 1 and 4 resulted in a 1 log_10_ reduction in *C. jejuni* load. However, no reduction was observed in the groups that received *L. salivarius*, *L. crispatus*, or *L. reuteri* [181]. In another study, oral daily administration of *L. gasseri* SBT2055 for 14 days significantly reduced *C. jejuni* colonization by up to 2.5 log_10_ [182]. Similarly, Abdelaziz et al. (2019) showed that *L. salivarius*, *L. johnsonii*, *L. crispatus*, and *L. gasseri* variably suppressed *C. jejuni* and downregulated *flaA*/*flaB*/*flhA*, *ciaB*, and *luxS* while enhancing chicken macrophage phagocytosis and activation, indicating potent anti-*Campylobacter* and immunomodulatory potential [183].

Due to the variability in outcomes with single-species probiotics, some studies demonstrated that incorporating multiple species may provide a more effective approach. For example, administering a mixture of five probiotic species comprising *L. salivarius*, *L. reuteri*, *E. faecium*, *Pediococcus acidilactici*, and *Bifidobacterium animalis* significantly reduced *C. jejuni* count by 6 log_10_ compared with the control group [184].

The inclusion of *Bacillus subtilis* C-3102 into the diet for broilers continuously for 42 days improved their body weight and decreased *Campylobacter* on the processed carcass by 0.2 log_10_ [185]. In two trials using Cobb 500 broilers, Aguiar and his group tested a mixture of 10 isolates of *Bacillus* spp. through oral and intracloacal delivery to day-old chicks, and their results revealed that *Bacillus* delivered orally reduced *C. jejuni* count by 1 log_10_, while those delivered intacloacally reduced the *C. jejuni* load by 1–3 log_10_ [186]. The inclusion of 2.5 × 10^6^ CFU/mL of *B. subtilis* PS-216 spores in drinking water from day 1 post-hatch until day 20 was also found to reduce *C. jejuni* by 1.2 log_10_ CFU/g [187]. Ismail and colleagues investigated whether the encapsulation of *B. amyloliquefaciens* into nanoparticles (BNPs) would boost its efficacy against *C. jejuni* in chickens, and their results revealed that supplementation of BNPs at a concentration of 7.5 × 10^5^ CFU/g to the feed from 1–35 days post-hatch resulted in significant fecal and cecal reduction at 7 days post-infection by 3.8 log_10_ and 3.9 log_10_, respectively [188]. In another study, including two *B. subtilis* C-3102 into the feed from day 1 until day 42 reduced *C. jejuni* colonization by 0.25 log_10_ on day 14 and 1.7 log_10_ on day 42 of age, while *B. amyloliquefaciens* CECT5940 reduced it by 1.12 log_10_ on day 14 and 1.2 log_10_ on day 42 [189]. In contrast to these findings, the inclusion of *B. subtilis* DSM17299 in the feed from day 21 until day 42 failed to significantly reduce the *C. jejuni* count in broiler chickens [190].

*Enterococcus faecium* is a gram-positive, anaerobic bacterium that beneficially colonizes the chicken gut and produces bacteriocin metabolites. A study conducted by Netherwood et al. (1999) showed that oral administration of *E. faecium* NCIMB 11508 strain at one and 28 days of age failed to reduce the relative abundance of *C. jejuni* in naturally infected chicks [191]. In another study, adding 10^9^ CFU/mL of *E. faecium* EM4 to drinking water for 21 days resulted in a 0.8 log_10_ reduction at 21 days post-administration and a 0.25 log_10_ reduction on day 35 (2 weeks after cessation of the probiotic) [192]. Dietary supplementation of *E. faecium* with *L. acidophilus*, *L. casei*, and *B. thermophilus* until 42 days post-hatch resulted in a reduction in *C. jejuni* abundance by 12% in naturally infected chickens [193]. Probiotic applications against *C. jejuni* in poultry are summarized in Table 7.

A non-pathogenic isolate of the *E. coli* family, *E. coli Nissle* (EcN), was isolated and characterized by Alferd Nissle in 1917 from human feces. Owing to its absence of virulence factors typically present in other pathogenic *E. coli* strains, such as invasiveness, enterotoxin or cytotoxin production [194,195], combined with its immune-modulatory properties, ability to enhance intestinal barrier function, and production of antimicrobial microcin peptides, EcN 1917 has been used as a probiotic to combat pathogenic bacteria, including *Salmonella*, *L. monocytogenes*, *Shigella* and *Campylobacter* [194,195]. In a recently conducted study by Helmy et al. (2022), chickens orally treated with free EcN in drinking water for two weeks starting at 3 weeks of age reduced *C. jejuni* cecal load by 2.0 log_10_ at week 5 of age [196]. The same group explored different administration routes, timing, and nanoparticle encapsulation of EcN, but observed no significant enhancement, as both encapsulated (9.8 × 10^8^ CFU/bird) and free EcN (1 × 10^9^ CFU/bird) reduced cecal *C. jejuni* by 2.0 and 2.5 log_10_, respectively, after oral administration three times weekly during weeks 4 and 5 of age [196].

**Table 7 microorganisms-13-02363-t007:** Probiotics’ effectiveness against *Campylobacter jejuni* in poultry.

Breed	Strain of Probiotics	Delivery	*Campylobacter* Strain/Dose	Effect on *Campylobacter* Colonization	References
Broiler	Single strain				
	*E. faecium* NCIMB 11508	First-day post-hatch and day 28 orally	Naturally infected	There is no reduction in the relative abundance of *Campylobacter*	[191]
	Calsporin^®^ (*B.* *subtilis* C-3102)	Day 1–42 in feed	Fecal contamination during processing	0.2 log_10_ reduction on chicken carcasses	[185]
	*Bacillus* spp. (10 isolates individually tested)	Per os and intracloacally at one day old	*C. jejuni* cocktail of 4 strains (2.5 × 10^6^ CFU)	Intracloacally: 1–3 log_10_ Orally: 1 log_10_ for only one isolate	[186]
	*L. salivarius* SMXD51	Given orally on day one and then every two to three days for 35 days	*C. jejuni* C97ANSES640 (1 × 10^4^ CFU)	0.8 log_10_ at 14 days and 2.81 log_10_ at day 35.	[178]
	*L. plantarum* PA18A	Orally, on days 1 and 4	*C. jejuni* strain 12/2 (1 × 10^4^ CFU)	1 log_10_ reduction	[181]
	*E. faecalis MB* 5259	Day 1–21 orally	*C. jejuni* MB 4185 (KC 40) (2 × 10^4^ CFU)	0.4 log_10_ in only one of the groups received 10^4^ CFU *E.* *faecalis* No reduction in the chickens received 10^8^ CFU *E. faecali*	[122]
	*E. coli Nissle* 1917 (free and chitosan micro-encapsulated)	Daily or three times per week supplementation in drinking water at weeks 4 and 5 of age	Cocktail of six *C.* *jejuni* strains/orally/(1 × 10^5^ CFU)	Up to 2.6 log_10_ at the end of the experiment at	[196]
	*B. amyloliquefaciens*-loaded nanoparticles (BNPs)	Per os from 1–35 post-hatch with three different doses of BNPs: I (2.5 × 10^5^ CFU/g), BNPs II (5 × 10^5^ CFU/g), and BNPs III 7.5× (CFU/g) of feed	Crop gavage with pandrug-resistant (PDR) and multi-virulent field *C. jejuni* 108 CFU/mL at 30 days old	BNPs III inclusion showed significant fecal and cecal reduction at 7 days post-infection (3.86 log_10_, 3.94 log_10_, respectively)	[188]
	*B. subtilis* PS-216 spores	2.5 × 10^6^ CFU/mL in drinking water 1–20 d	8 d, all of the broilers were inoculated with 4 × 10^6^ CFU *C. jejuni* 11,168 by oral gavage	1.2 log_10_ CFU/g feces in the *C. jejuni* counts	[187]
	*L. plantarum* 256	(10^7^ CFU/mL) in drinking water for 6 and 9 weeks *L. plantarum* strain 256 during baling, providing an inoculum concentration of 10^8^ CFU per gram of fresh matter	10^6^ CFU/mL of the *C. jejuni* strain 65 at day 22 for the (6 weeks exp) and at 29 days for the (9 weeks exp)	No significant reduction at the end of the experiments at 42 and 63 days.	[197]
	Mixed strains				
	K-bacteria + competitive exclusion Broilact^®^	Day 1–38 in drinking water	*C. jejuni* T23/42 (1.3 × 10^4^ CFU)	Up to 2 log_10_	[140]
	*Citrobacter diversus* 22 + *K. pneumonia* 23 + *E. coli* 25 + mannose	Days 1 and 3	*C. jejuni* orally (10^8^ CFU)	Up to 70% reduction	[198]
	Avian Pac Soluble (*L. acidophilus* + *Streptococcus faecium*)	Day 1–3 in drinking water	*C. jejuni* C101 (2.7 × 10^4^ CFU)	Two-thirds reduction in *C.* *jejuni* shedding	[199]
	PrimaLac (*L.* *acidophilus* + *L.* *casei + B.* *thermophilus +* *E. faecium*)	Day 1–42 in feed	Naturally infected	12% reduction of *C. jejuni* presence	[193]
	*B. longum* PCB 133	Day 1–15 intraesophageally	Naturally infected	1 log_10_ reduction	[200]
	Microencapsulated *B. longum* PCB133 + oligosaccharides	Day 1–14 in feed	Naturally infected	Up to 1.4 log_10_	[201]
	PoultryStar sol^®^ (*E. faecium* + *P. acidilactici* + *B. animalis* + *L. salivarius* + *L. reuteri*)	Day 1–15 in drinking water	*C. jejuni* 3015/2010 (10^4^ CFU)	6 log_10_	[184]
	*L. acidophilus* NCFM or *L. crispatus* JCM5810 or *L.s gallinarum* ATCC or *L. helveticus* CNRZ32	Day 1 and 4 orally	*C. jejuni* F38011 (10^8^ CFU)	~2 log_10_ reduction	[202]
	*L. gasseri* SBT2055 LG2055 WTCM, Dapf1 and Dapf2 mutant strains	Day 2–14 orally Dapf1: No reduction	*C. jejuni* 81–176 (10^6^ CFU)	WTCM and Dapf2: Up to 270-fold reduction	[182]
	*Bacillus* spp.+ *L. salivarius* subsp. *salivarius + L.* *salivarius* subsp. *salicinius*	Day 1 orally	*C. jejuni* cocktail of 4 strains (2.5 × 10^6^ CFU)	1–2 log_10_ in only one of the three trials	[203]
	*L. paracasei J.R +* *L. rhamnosus* *15b + L. lactis Y +* *L. lactis FOa*	Day 1–42 in drinking water	Naturally infected	Up to 5 log_10_	[204]
	Calsporin^®^ (*B.* *subtilis* C-3102) Ecobiol^®^ (*B.* *amyloliquefaciens* CECT 5940)	Day 1 and 42 in feed	*C. jejuni* C97ANSES640 (10^4^ CFU)	Calsporin^®^: 0.25 log_10_ reduction on day 14 and 1.7 log_10_ on day 42 Ecobiol^®^: 1.12 log_10_ on day 35 and 1.2 log_10_ on day 42	[189]
	*B. subtilis* DSM 17299 or *S. cerevisiae* *boulardii*	Day 21–42 in feed	*C. jejuni* ST45 (10^4^ CFU)	*B. subtilis*: No reduction S. *cerevisiae*: Up to 0.3 log_10_ reduction	[190]
	Lavipan (multispecies probiotic): *L.* *lactis* IBB 500, *Carnobacterium divergens* S-1, *L. casei* OCK 0915, L0915, *L.* *plantarum* OCK 0862, and *S. cerevisiae* OCK 0141	Day 1–37 in feed	Naturally infected	<1 log_10_	[122]
Layers	*Citrobacter diversus*, *K.* *pneumoniae*, and *E. coli*			Reduced *C. jejuni* load in ceca	[198]
	Cecal culture			Reduced *C. jejuni* load in ceca	[205]
	*E. faecium* EM41	Orally in drinking water, 10^9^ CFU/mL were received for 21 days	Natural infection	0.8 log_10_ reduction at 21 days of starting administration and 0.25 log_10_ reduction at day 35 (2 weeks after cessation of the additive)	[192]
	Enterocin EM41 (Ent EM41)	Enterocin (Ent) EM41 (40 μL/animal/day, 25,600 AU/mL).	Natural infection	1.95 log_10_ reduction at 21 days of starting administration and 0.75 log_10_ reduction on day 35 (2 weeks after cessation of the additive)	[192]
Duck	*L. salivarius*	Orally in feed 2 × 10^8^ CFU/g for 79 days	Natural infection	No reduction	[179]
Turkeys	Bacteriocin of *P. polymyxa* and *L.* *salivarius*	Three successive days on 10–12 post-hatch	Orally, 10^6^ CFU of a mixture of 3 *Campylobacter coli* isolates.	4 log_10_ reduction in *Campylobacter* concentrations	[180]

### 2.3. Clostridium Perfingens

*C. perfringens* is a Gram-positive, spore-forming, toxin-producing anaerobic bacterium that inhibits the poultry gut [206]. As a foodborne pathogen, *C. perfringens* has been associated with food poisoning, gas gangrene, and diarrhea in humans and NE in poultry [207]. Epidemiologically, *C. perfringens* carrying the enterotoxin gene (*cpe*) is ranked as the second leading cause of foodborne bacterial illness in the United States, accounting for roughly one million incident cases annually and $400 million USD in annual economic losses. Although most infections are self-limiting, they can necessitate hospitalization in vulnerable populations, incurring high healthcare costs.

#### 2.3.1. *C. perfringens* Virulence, Pathogenicity and Mode of Transmission

Pathogenic strains of *C. perfringens* can produce over 20 toxins, with six primary toxins being alpha, beta, epsilon, iota, enterotoxin, and necrotic B-like (Net B) toxin. These six toxins are encoded by the genes *cpa/plc*, *cpb*, *etx*, *iap/iab/itx*, *cpe*, and *netB*, respectively. The composition of these specific genes determines the classification of different *C. perfringens* strains, categorized as toxinotypes, depicted in Table 8. Rood et al. (2018) expanded the scheme to include toxinotypes F and G to compensate for the increasing complexity and diagnostic limitations of the outdated framework [208].

*C. perfringens* produces a diverse range of toxins integral to its pathogenicity in humans and animals. Among these, *C. perfringens*’ CPE, associated with the type F strain, is one of the most clinically relevant toxins in human foodborne infections. CPE binds to claudin receptors on intestinal epithelial cells, disrupting tight junctions and inducing pore formation [209]. This pore formation increases intestinal permeability, fluid accumulation, and epithelial necrosis, leading to symptoms like diarrhea and abdominal cramps commonly seen in foodborne outbreaks.

In poultry, *C. perfringens*’ type G strain produces NetB toxin, the primary toxin responsible for NE in chickens. NetB forms heptameric pores in the membranes of intestinal epithelial cells, leading to cellular disruption, necrosis, and mucosal damage. It shares structural homology with other pore-forming toxins like beta toxin and *Staphylococcus aureus* α-hemolysin. Although its receptor has not been definitively identified, its cation-selective pore formation is integral to its pathogenicity in poultry [210]. There is also some suggestion that carriage of *tpeL* in some *netB+* *C. perfringens* strains may contribute to enhanced virulence in chickens [211,212,213]. NE creates economic losses exceeding $6 billion USD annually for poultry producers due to performance reductions, higher feed conversion ratios, and treatment costs [214]. NE can present acute high mortality rates of 10–40% among two to six-week-old broilers or sub-clinically affect broilers, compromising feed conversion and growth rates [215].

The concern surrounding *C. perfringens* stems from the zoonotic potential of specific bacterial strains. For example, a strain carrying the *netB* gene may induce NE in poultry under conditions, such as contaminated meat products and subsequent human ingestion. Once consumed, the bacterium may express the *cpe* gene, enabling foodborne illness in humans [206]. The toxin-producing capability of *C. perfringens* imposes a significant threat to human and poultry health in the US, contributing to one million foodborne illness cases in addition to substantial economic losses within the poultry industry [15].

#### 2.3.2. Probiotic Efficacy Against *C. perfringens*

A review by Kulkarni et al. (2022) highlighted that probiotics, across genera such as *Lactobacillus*, *Enterococcus*, *Bacillus*, *Bacteroides*, and some yeasts, consistently reduced *C. perfringens* colonization and NE–related pathology in chickens, while enhancing bird performance [18]. Multiple studies evaluated the effectiveness of mono-strain probiotics in reducing *C. perfringens* burden in broiler chickens, depicted in Table 9. Granstad et al. (2020) investigated three individual strains: *B. subtilis* (PB6), *B. subtilis* (no. 671265), and *L. farciminis*. *B. subtilis* synthesizes bacteriocins such as mersacidin and sublancin that disrupt bacterial cell membranes and inhibit cell wall biosynthesis to suppress *C. perfringens* proliferation [216]. Among these strains in comparison to the control group, *B. subtilis* strain PB6 led to the highest *C. perfringens* log_10_ reduction of 0.98, whereas its other strain only exhibited a 0.19 log reduction, and *L. farciminis* gave a 0.63 log_10_ reduction [216]. However, even with such variability among log reduction counts, each probiotic strain reduced FCR and increased the BW of the broilers. Similarly, Gharib-Naseri et al. (2021) and Wu et al. (2018) utilized *Bacillus*-based probiotics, *B. amyloliquefaciens* and *B. coagulans*, respectively, to investigate their roles in gut microbiota [217,218]. It was found that *B. amyloliquefaciens* exhibited 0.8 log_10_ reduction of cecal *C. perfringens*, whereas *B. coagulans* showed log reductions of 0.84, 1.46, and 1.79 at days 28, 35, and 42 of age, respectively. *B. coagulans* enhanced mucosal immunity by the production of secretory immunoglobulin A and the reduction of systemic immunoglobulin G levels, while also upregulating antimicrobial peptides, such as folicidin-2 and lysozyme, to suppress *C. perfringens*. Focusing on the outcomes of Wu et al. (2018), the levels of *C. perfringens* reductions and growth performance varied, where significant increases in BW and improved FCR were found from day 15 to 21 when supplemented with dietary *B. coagulans* [217]. Overall, single-strain probiotics can improve broiler health and performance, but can express variability in *C. perfringens* reductions, depending on the probiotic strain used [217].

Multi-strain probiotic formulations demonstrated greater outcomes through synergistic mechanisms throughout diverse bacterial species. Granstad et al. (2020) created a multi-strain probiotic feed additive using *E. faecium*, *B. animalis*, and *L. salivarius* to determine its impact on broilers [216]. The multi-strain probiotic showed a 0.88 log_10_ reduction in cecal *C. perfringens* with improvements in BW and FCR compared to the control birds [216]. Abd El-Ghany et al. (2022) also developed a multi-strain probiotic comprised of *B. subtilis* and *B. licheniformis*, administered through the feed, and observed a 2.45 log_10_ reduction in cecal *C. perfringens* with reductions in *C. perfringens*-related mortality rates and lower FCR values [219]. These strains utilized in tandem contribute to the inhibition of *C. perfringens* growth and toxin production by enhancing mucosal immune responses. Compared with earlier studies, McReynolds et al. (2009) designed a multi-strain probiotic as a water additive rather than a feed additive [220]. The probiotic blend colonized the gut and acidified the environment while producing bacteriocins such as reuterin and hydrogen peroxide to deter *C. perfringens* establishment, thus resulting in a 2.91 log_10_ reduction of *C. perfringens* in the cecal content of broiler chickens [220]. In addition, this probiotic mix showed reductions in intestinal lesions and mortalities compared to the positive controls. The multi-strain probiotic combinations provide many benefits for gut microbiota stability, nutrient absorption, and immune support. However, the strain-specific nature of probiotic efficacy requires precise selections where not all mixes guarantee a significant outcome. Buiatte et al. (2023) performed an in vitro study on a *Bacillus*-based probiotic mix comprising *B. subtilis*, *B. licheniformis*, *B. coagulans*, and *B. pumilus* [221]. Inter-strain antagonism was noted when the probiotic mix had a 0.28 log_10_ reduction in cecal *C. perfringens*, whereas *B. subtilis* alone induced a 6 log_10_ reduction [221]. This difference in bacterial counts suggests that competitive interactions or bacteriocin inhibitors may compromise the overall effectiveness of a multi-strain probiotic formulation.

**Table 9 microorganisms-13-02363-t009:** Probiotics’ effectiveness against *C. perfringens* in broilers.

Probiotic Strain(s)	Administration	Concentration	Main Outcomes	References
*B. amyloliquefaciens*	Feed additive	10^6^ CFU/g feed	0.8 log_10_ reduction in cecal *C. perfringens* counts. Improved FCR and BWG	[218]
*B. coagulans*	Feed additive	4 × 10^9^ CFU/kg feed	0.84, 1.46, and 1.79-log_10_ reductions in *C. perfringens* cecal counts at days 28, 35, and 42, respectively. Decreased lesion scores and reduced crypt depths in the small intestine	[217]
*B. subtilis*	Feed additive	2 × 10^8^ CFU/g feed	0.98 log_10_ reduction in cecal *C. perfringens* counts. Improved FCR and BWG	[216]
*B. subtilis*	In vitro	10^8^ CFU/mL	6 log_10_ *C. perfringens* reduction alone. Efficacy declined when combined with other *Bacillus* strains	[221]
*E. faecium*, *B. animalis*, and *L. salivarius* mix	Feed additive	2 × 10^8^ CFU/g feed	0.88 log_10_ *C. perfringens* count reduction. Enhanced BWG and FCR	[216]
*B. subtilis* and *B. licheniformis* mix	Feed additive	2.5 × 10^12^ CFU/kg feed	2.45 log_10_ *C. perfringens* reduction. Reduced mortality and lesion scores	[219]
*E. faecium*, *B. animalis*, *P. acidilactici* and *L. reuteri* mix	Water additive	1 × 10^9^ CFU/mL	2.91 log_10_ *C. perfringens* reduction. Reduced lesion scores and mortalities.	[220]

Overall, current findings emphasize the need for precision strain selection and optimized dosing strategies. The route and consistency of administration, viability of spores during feed processing, and age-specific gut microbiota dynamics in broilers are all critical factors determining probiotic success. While individual strains like B. subtilis offer impressive monotherapeutic potential, the growing body of evidence supports the synergistic use of tailored multi-strain or synbiotic formulations, especially in antibiotic-free production systems.

Safety evaluations still need to precede widespread synbiotic and multi-strain probiotics applications. Concerns regarding horizontal gene transfer of antibiotic resistance markers from probiotic strains remain valid; therefore, future applications should adopt genomic screening and resistance profiling practices to ensure regulatory compliance and long-term safety for all parties involved.

### 2.4. Escherichia coli

*E. coli* is a gram-negative bacterium commonly present in controlled levels within the digestive tracts of chickens. While many strains of *E. coli* are nonpathogenic, pathogenic strains such as *E. coli* O157, K88, and O78 can cause severe illness in broilers [222,223,224,225]. APEC, particularly strain O78, is the primary cause of colibacillosis in broilers, which is reported to cause a mortality rate of 6.56% in infected chickens, with clinical signs of inflammation, yolk sac infections, coligranulomas, swollen head syndrome, avian cellulitis, and enteritis [226]. It has also been estimated that about 30% of broilers are sub-clinically infected with *E. coli* at a given time [227]. Additionally, the presence of pathogenic strains of *E. coli* poses a zoonotic threat as they can spread through the consumption of contaminated poultry meat and have also been detected on the shells and in the contents of eggs [228,229].

The economic costs of *E. coli* both in the US and globally are associated with both the poultry industry and the healthcare industry. Monetary losses in the poultry industry are often due to containment of the disease and mortality of the birds, as well as decreased weight gain and production inefficiencies [230,231]. Additionally, Shiga-toxin-producing *E. coli* was one of the top three causes for total food recall by the USDA Food Safety and Inspection Service between 1994 and 2015 [232]. A 2019 estimate found that the O157 strain of *E. coli* alone has accrued economic losses of about $268.3 million, while other strains cost about $7.7 million [233].

#### 2.4.1. *E. coli* Virulence, Pathogenicity and Mode of Transmission

Different strains of *E. coli* exhibit different virulence factors to cause disease in their hosts, which are primarily mammals and birds. Pathogenic strains of *E. coli* are identified based on their surface antigen, which is either of the O, K, or H variety. Adherence of the bacteria to the colonization surface is often accomplished through adhesion proteins and outer membrane proteins. They have also been known to create biofilms and resistance to the immune complement system, which can increase pathogenicity [234]. The virulence factors associated with pathogenic *E. coli* strains are provided in Table 10.

*E. coli* infection in the poultry was previously controlled through the practice of using antibiotics at subtherapeutic levels in the diet [231]. Antimicrobial resistance in *E. coli* is a growing concern due to the bacterium’s capacity to act as a reservoir and vector for resistance genes, with studies linking poultry-origin *E. coli* strains to urinary tract infections in humans [231]. This has also caused many strains to become resistant to ampicillin, tetracycline and gentamicin [226]. Quinolone-resistant *E. coli* strains have been detected in poultry carcasses and increasingly among human clinical isolates in countries like Spain and Taiwan, emphasizing direct foodborne transmission.

#### 2.4.2. Probiotic Efficacy Against *E. coli*

Probiotics have been used as a safer alternative for controlling *E. coli* in broilers. *Lactobacillus*, a non-spore-forming, gram-positive bacterium naturally found in the gut, is widely used as a probiotic due to its safety [224]. However, its effectiveness against *E. coli* in broilers has shown variable results. *L. plantarum* has been found to withstand low pH during in vitro studies and survive many different bile level concentrations between 1–3%, causing inhibition of *E. coli* O157 growth in broilers [243]. Dietary supplementation of *L. plantarum* showed an increase in IgA secretion and a reduction in *E. coli* counts in cecal digesta by 0.69 log_10_ at day 42 of age [244]. In a subsequent study, chicks supplemented with *L. plantarum* and challenged with *E. coli* K88 had an increase in ileal mucosal IgA and a reduction in *E. coli* in the cecal digesta of 0.3 log_10_. Additionally, decreased ileal mRNA expressions of IL-2, INF-γ, TNF- α, IL-4 and TLR4 were demonstrated [223]. Contrastingly, another study showed that healthy Ross 308 chicks supplemented with 1 × 10^6^ to 1 × 10^8^ CFUs of *L. plantarum*/mL of drinking water had no significant reduction in cecal *E. coli* colonization, and this was attributed to its inability to colonize the mucosal wall of the intestines [245].

The use of a commercial *Lactobacillus* spp. mixture containing 2 strains of *L. acidophilus*, 3 strains of *L. fermentum*, 1 strain of *L. crispatus*, and 6 strains of *L. brevis* was studied for probiotic tendencies. It was found that *Lactobacilli* strains effectively reduced naturally occurring *E. coli* counts in broiler chickens, with one trial showing a reduction of 1.83 log_10_ and the other showing complete inhibition [246]. Another similar experiment found that supplementing *Lactobacillus* significantly reduced *E. coli* colony counts, with a decrease of 0.57 log_10_ CFUs/g of excreta [224].

The heat-resistant property of *B. subtilis* allows it to survive feed preparation processes [247]. When *B. subtilis* was used as a dietary supplement in Ross 308 chicks challenged with *E. coli*, birds showed a reduction in *E. coli* colonization by 1.69 log_10_ CFUs/g. Another study reported a complete inhibition of *E. coli* in the liver and spleen 48 h post challenge, and a reduction of approximately 3.2 × 10^7^ CFUs/g in the cecum [248]. *E. faecalis* is a potential commensal probiotic, but has shown pathogenicity when it gains antibiotic resistance, sensitivity to the acidic environment and toxin release into the body [249,250]. When *E. faecalis* was supplemented in the drinking water to *E. coli* O78 challenged broilers, no change was found between bacterial or coliform counts, but a significant increase in serum IgY levels was observed [225]. Dong et al. (2019) showed that broilers challenged with *E. coli* K88 and then given microencapsulated *E. faecalis* as a dietary supplement had greater serum IgA levels and a reduction in cecal *E. coli* colonization by approximately 0.5 log_10_ CFU/g [250]. Encapsulation may protect the probiotic from changes in the external environment, such as food processing or harsh conditions in the gastrointestinal tract, allowing it to survive longer and have its effect. Huang et al. (2019) have also studied *E. faecium* as a potential probiotic candidate by supplementing it in the diet and challenging broilers with *E. coli* O78 [251]. Supplemented birds maintained higher levels of claudin-1 and occludin mRNA expression, along with a reduction in *E. coli* colonization by approximately 2.0 log_10_ CFUs/g in the liver [251]. A subsequent study challenged male Cobb broilers with *E. coli* K88 and showed that supplementation with *E. faecium* increased levels of TNF-α and serum IgA, and there was a reduction of *E. coli* colonization in the cecum by 0.27 log_10_ CFU/g [252].

Another probiotic candidate, *Clostridium butyricum*, when used as a supplement in broilers that were challenged with *E. coli* K88, showed increased levels of TNF-ɑ and IL-4 [253]. Subsequently, another study found that broilers orally challenged with *E. coli* K88 and given *C. butyricum* had reduced cecal *E. coli* colony counts of approximately 1 log_10_ CFUs/g. The birds also had increased serum IgA, IgY, and IgM levels on day 3 post-challenge, and increased IgA and IgM levels on day 14 post-challenge.

*S. cerevisiae* has been used as a probiotic within the poultry industry in the past. Igbafe et al. (2020) tested the antagonistic effect of *S. cerevisiae* in vitro as a supplement against *E. coli* O157, with the results showing its ability to withstand low pH values; however, it showed no antagonistic effects when plated with *E. coli* [254]. Further confirming this finding, an in vivo study by Bortoluzzi et al. (2018) using healthy male Ross 308 chicks fed a basal diet with SINERGIS, produced by Aleris Nutrition (Aleris USA LLC, NV), a commercially available form of *S. cerevisiae*, found no significant reduction of *E. coli* colonization in the ileum or cecum [255].

*Candida famata*, another species of yeast, was used in drinking water at 10^8^ CFUs/mL, and it was found that there was no significant reduction in cecal *E. coli* colonization. It is believed that while this species had increased proliferation of many other yeast species within the bird’s intestines, it did not have any effect on cecal *E. coli* colonization [245].

Similarly, another probiotic candidate, *Lacticaseibacillus rhamnosus*, has been found to have inhibitory effects by producing peptides against APEC. Guo et al. (2021) studied the efficacy of *L. rhamnosus* against pathogenic *E. coli* O78 in Leghorn chickens and found a 1.6 log_10_ CFU reduction in cecal *E. coli* colonization, with no colonization in the heart and liver [256]. Another study found that *L. rhamnosus* can act as a competitive excluder against *E. coli* and prevent its adhesion to the intestinal epithelia of chickens [257]. These supplemented birds also had an upregulation of serum IgA, IgG, and IgM when measured on day 21 of the experiment, and upregulation of proinflammatory cytokines, IFN-α, IL-1β, and IL-6 in the spleen. *E. coli*-challenged birds supplemented with *L. rhamnosus* had a significant reduction of *E. coli* colonization in the heart, lungs, and kidney by 1.5 log_10_ CFUs/g, 1.0 log_10_ CFUs/g, and 0.5 log_10_ CFUs/g, respectively, on day 3 post-challenge [256].

*Bifidobacterium lactis* has been used in the food industry, specifically the dairy industry, as a probiotic in milk and yogurt [258,259]. It has been found to inhibit colonization of APEC strains in vitro. Kathayat et al. (2022) studied the efficacy of *B. lactis* in controlling levels of pathogenic *E. coli* O78 in Leghorn chickens and found that birds treated with *B. lactis* had a 0.7 log_10_ CFU reduction in *E. coli* colonization in the cecal contents, while 10% of chickens were positive for *E. coli* in the liver, and 20% of chickens were positive for *E. coli* in the heart [257]. Table 11 summarizes the effectiveness of probiotics against *E. coli* serotypes in broiler chickens.

### 2.5. Listeria Monocytogenes

*L. monocytogenes* is a Gram-positive, rod-shaped, non-spore-forming, facultative anaerobic psychrophilic bacterium responsible for listeriosis. Although there are 21 species of *Listeria*, only *L. monocytogenes* and *L. ivanovii* have been considered pathogenic, with *L. monocytogenes* being the most important. The CDC estimates approximately 1600 annual cases in the U.S., with a mortality rate of 17.6% in high-risk groups.

*L. monocytogenes* poses significant risks to public health due to its ability to thrive in harsh conditions, including refrigeration temperatures, acidic environments (pH 2.0–9.6), and high osmolarity, enabling persistence in food processing plants [268]. The ability of *L. monocytogenes* to produce biofilms also limits the efficacy of disinfectants and antimicrobial agents. It can survive various processing conditions, including freezing, surface dehydration, and simulated spray chilling. It can also survive in vacuum-packaged chicken and other ready-to-eat (RTE) foods.

It is ubiquitous in nature, found in soil, water, and animal feces, and frequently contaminates raw and RTE foods such as poultry, dairy products, fresh vegetables, seafood, and processed meats. The pathogen is particularly dangerous for vulnerable populations, including pregnant women, neonates, the elderly, and immunocompromised individuals, with mortality rates exceeding 20% in these groups [269,270]. Neonatal infections can occur through maternal chorioamnionitis (early-onset sepsis) or birth canal exposure (late-onset meningitis), while immunocompromised adults are at risk of septicemia, meningitis, and rhombencephalitis [271,272] as it can breach the blood-brain barrier. Furthermore, viral pathogens associated with gastroenteritis may allow for *L. monocytogenes* translocation.

While sporadic in poultry, it primarily affects young chicks, causing septicemia or localized encephalitis. It often occurs alongside other infections like coccidiosis or salmonellosis, highlighting its opportunistic nature [273]. *L. monocytogenes* exhibits varying prevalence, with broilers (32%) showing higher contamination rates than laying hens (15.5%), and free-range flocks (37%) more affected than conventional systems (28%) [274]. While clinical disease is rare in adult poultry, young chicks are highly susceptible, suffering septicemia or encephalitis with mortality rates up to 40% [270,275].

#### 2.5.1. Listeria Virulence, Pathogenicity and Mode of Transmission

Whole-genome sequencing (WGS) analysis of 90 *L. monocytogenes* isolates revealed that clonal complex CC9 was the most prevalent in poultry and livestock meat and shared 80 other resistance genes. Poultry meat isolates had a higher number of tetracycline resistance genes (*tetZ*, *tet41*, and *tetA*), virulence genes, such as *inlA* (which promotes host cell invasion) and *bsh* (which helps resist bile toxicity), compared to livestock meat isolates [276].

The bacterium contaminates eggs through soil, poultry feces, or processing equipment, with eggshells showing a 1.8–6% prevalence [269]. Conventional caged systems have lower contamination rates (1.6%) due to controlled environments, but small-scale farms (<3000 hens) often escape regulatory oversight, increasing contamination risks [269]. Processing plants are critical contamination sites, with *Listeria* persisting in biofilms on equipment like egg washers, conveyors, and floor drains. Humans typically contract listeriosis from contaminated raw poultry meat or unhygienic processing conditions, rather than direct bird contact [277]. *L. monocytogenes* can colonize broiler chickens, but not as easily as other bacterial foodborne pathogens like *Salmonella* or *C. jejuni*. Younger birds are more susceptible to colonization than older birds, and the colonization response is dose-dependent. Surveys show contamination rates ranging from 12% (RTE) to 60% (fresh packaged chicken) [268]. Human listeriosis primarily results from consuming contaminated foods, with infective doses as low as 10^5^ CFUs in high-risk individuals and 10^7^–10^9^ CFU in healthy adults, leading to gastrointestinal symptoms [270,278].

*Listeria*’s virulence is linked to its capability of adhesion, invasion, and translocation across the gut barrier. Specific genetic lineages (e.g., CC1, CC4) and virulence factors, including internalins (InlA, InlB), listeriolysin O (LLO), and ActA, facilitate host cell invasion, intracellular survival, and cell-to-cell spread. Alternative sigma factor σB, a protein subunit of RNA polymerase (RNAP), aids in the ability of *Listeria* to survive in harsher conditions. Serotypes 1/2a, 1/2b, and 4b cause over 96% of human cases, with serotype 4b associated with severe outbreaks [270,279]. Major outbreaks, such as the 2016–2017 UK cooked chicken incident (linked to recalled products with 20–340 CFU/g) and the 2008 Canadian deli meat outbreak, highlight its public health and economic impact, with U.S. illness costs estimated at $186 million annually [280]. *L. monocytogenes* is the most serious zoonotic disease, with high rates of hospitalization (92%) and mortality (17.6%) [281]

Control strategies include stringent cooking protocols (e.g., 90 min at 65 °C) to overcome *Listeria*’s heat resistance, which exceeds that of *Salmonella* and *Campylobacter* [275]. Although no commercial probiotics are currently available to control *Listeria* infections, several research groups have investigated the potential of probiotic strains as natural bio-preservatives and antimicrobial agents against *L. monocytogenes*. Deng et al. (2020) showed that dietary supplementation of broilers with an equal blend of *L. acidophilus* and *L. plantarum* at doses of 10^6^, 10^8^ and 10^10^ CFU/kg significantly reduced *L. monocytogenes* loads in the cecum, liver, spleen, and skin by 0.065–0.9 log_10_ CFU compared to control groups [176]. In the intestine, it lowered proinflammatory cytokines (IL-1β, IL-6, TNF-α, and IFN-γ) by 25.4–51.1%, upregulated anti-inflammatory genes (IL-10, HIF1A, PTGER2, and PTGS2), and downregulated critical virulence genes responsible for adhesion (*Ami*, *FlaA*), invasion (*InlA*, *InlB*), and cytotoxicity (*HlyA*) of *L. monocytogenes*. Apart from this study, no other experimental or field studies specifically testing probiotics (or synbiotics) against *Listeria* in poultry were identified.

#### 2.5.2. Probiotic Efficacy Against Listeriosis

A couple of in vitro trials assess LAB as potential poultry probiotics against *L. monocytogenes*. Reuben et al. (2020) demonstrated potent antimicrobial activity of LAB with inhibition zones of 14–20 mm and high co-aggregation capability (up to 83.6%) [282]. Key probiotic candidates included *L. reuteri*, *P. acidilactici*, *P. pentosaceus*, and *E. faecium*, which exhibited acid and bile tolerance (surviving pH 2.0 and 0.3% bile salts), phenol resistance (withstanding 0.4% phenol), and strong adhesion to intestinal epithelial cells (3.0–6.0 log_10_ CFU/mL) and showed anti-*Listeria* activity consistent with pediocin/reuterin production and organic-acid–mediated inhibition. While these in vitro results are promising for controlling *Listeria* in poultry, further in vivo studies are needed to validate their efficacy under field conditions.

A study by Abouloifa et al. (2022) demonstrated the effectiveness of *L. plantarum* S61 in controlling *L. monocytogenes* in minced poultry meat [283]. The cell-free supernatant of this strain exhibited potent antibacterial activity, reducing *Listeria* counts from 5 log CFU/g to 2 log_10_ CFU/g after seven days of refrigerated storage. The antimicrobial effect was attributed to proteinaceous compounds, as treatment with protease enzymes neutralized the activity [283]. It also improved the meat’s physicochemical and sensory qualities, like color stability and delaying spoilage caused by other microbes. Additionally, biofilm-forming *L. plantarum* Y42 has shown high adhesion capacity to intestinal epithelial cells and effectively inhibited *L. monocytogenes* adhesion and invasion through CE. Its extracellular polymeric substances enhanced its ability to scavenge pre-formed *Listeria* biofilms more effectively than planktonic cells, suggesting an advantage in food preservation [284].

Regulatory agencies enforce strict microbiological criteria, requiring <100 CFU/g in non-growth-supporting foods [285]. Despite these measures, *L. monocytogenes* remains a persistent threat, necessitating improved surveillance, WGS for outbreak tracing, and innovative solutions to mitigate its impact on food safety [280].

## 3. Conclusions and Future Prospects

Bacterial foodborne pathogens such as *Salmonella* spp., *Campylobacter* spp., *L. monocytogenes*, *C. perfringens*, and pathogenic *E. coli* compromise poultry health and productivity and pose public health and economic burdens. This review brings together diverse evidence on the potential of probiotics as preharvest interventions to control these major bacterial foodborne pathogens in poultry. The increasing interest in probiotics for poultry production and research is driven by their ability to modulate gut microbiota, enhance gut immunity, and competitively exclude pathogens, while improving bird performance. Overall, although some studies show that probiotics can reduce intestinal colonization and shedding of these bacterial pathogens, their effects vary widely depending on strain, dose, formulation, bird age and type, administration route, treatment duration, sample size, and study design. For *Campylobacter*, *Lactobacillus* supplementation has been shown in some studies to reduce cecal loads by ~1–3 log_10_, while others report no effect; benefits sometimes require repeated dosing or multi-strain combinations. Similarly, *Bacillus* and LAB supplements can reduce cecal or fecal *Salmonella* loads by 0.4–3 log units, but some studies show minimal impact. For *C. perfringens*, single strains may be effective, though multi-strain mixes can either outperform or underperform due to inter-strain interactions, highlighting a design gap. For *Listeria*, in vivo poultry data are limited, and the lack of field-relevant trials represents a clear research gap.

A general observation is that many research trials use limited bird numbers and rearing conditions that differ from commercial settings, often lacking factors such as co-infections, litter reuse, diet variability, stocking density, vaccination, or litter microbiota interactions that influence pathogen dynamics. Studies also vary widely in their endpoints, ranging from cecal CFU and fecal shedding to carcass contamination, making comparisons between studies difficult. Formulation is another gap, as bacterial survival through pelleting, storage, and digestion is rarely verified. Most pathogen challenges use a single strain, whereas field flocks face diverse, multidrug-resistant serovars and mixed exposures, leaving cross-strain protection untested. Furthermore, it is also important to emphasize that supplementing poultry with individual probiotic species or a limited cocktail of a few strains may be insufficient to dominate the gut microbiota and confer the intended benefits, potentially allowing pathogenic bacteria to colonize. Safety and quality standards also need strengthening through consistent genomic screening for resistance genes, toxins, potency, and stability. Finally, integration with other interventions such as vaccines and synbiotics shows potential. By integrating findings on strain efficacy, delivery strategies, and pathogen outcomes, this review highlights how probiotics can effectively reduce foodborne risks while promoting bird health and performance.

With recent advancements in the multi-omics tools, microbiome profiling can now be performed at the species level, enabling the identification of numerous beneficial microbes that could not be detected using traditional culture-based methods. Functional characterization of these microbes through metagenomic and metabolomic approaches can facilitate the development of a consortium of well-defined, well-characterized probiotic strains. Such next-generation poultry-specific probiotics could establish a stable and synergistic microbiome in neonatal chicks, thereby enhancing microbial interactions and minimizing colonization by pathogenic bacteria.

## Figures and Tables

**Figure 1 microorganisms-13-02363-f001:**
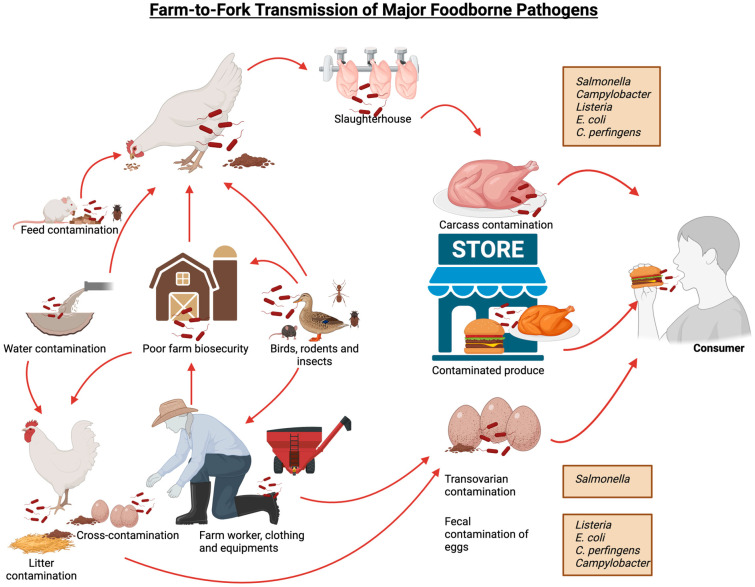
Farm-to-fork transmission pathways of different bacterial foodborne pathogens.

**Table 1 microorganisms-13-02363-t001:** Based on their host specificity, *Salmonella* serovars are classified into the following three groups.

Group	Host	Serovars	Infections	References
Group I (Host restricted)	Humans and higher primates	*S.* Typhi, *S.* Paratyphi *A*, *B*, *C*, and *S.* Sendai	Typhus, diarrhea, septicemia, and abortion in mares. Serovars such as *S.* Paratyphi A, *S.* Paratyphi B, *S.* Paratyphi C, and *S.* Sendai are the causal agents of typhoid fever.	[51,52]
Group II (Host)	Specific animal hosts	*S.* Dublin in cattle, *S.* Gallinarum in poultry, *S.* Abortusequi in horses, *S.* Abortusovis in sheep, and *S.* Choleraesuis in pigs	Septicemia, enterocolitis in cattle, fatal systemic infection in swine, bacteremia in humans and mouse. Causes subclinical infection in later hosts, making them a reservoir and carrier of the pathogen.	[51,52,53]
Group III (Unrestricted serovars)	Wide host range, including humans, animals, and the environment	*S.* Typhimurium and *S.* Enteritidis	Cause relatively less severe enteric diseases as compared to the serovars from Group I and Group II. Lack mechanisms for invading the mature immune system of older hosts, therefore causing more severe diseases in young hosts. Enterocolitis in humans and swine, asymptomatic carriers in poultry and cattle, and septicemia in mouse.	[52,54]

**Table 2 microorganisms-13-02363-t002:** SPIs: Regions on the *Salmonella* plasmid encoding several virulence factors.

*SPI* Type	Function	References
*SPI-1*	Host cell invasion and macrophage apoptosis induction. Possesses translocons involved in *Salmonella* contact and invasion and colonization of mammalian epithelial cells. T3SS encoded by *SPI-1* functions for suppressing early proinflammatory cytokine expression in macrophages, including that of IL-1β, IL-8, TNF-α, IL-23α, GM-CSF, and IL-18. MHC II downregulation and polarization to the M2 phenotype in macrophages increases the blood-brain barrier penetration along with protein A genes.	[67,68,69]
*SPI-2*	Proliferation within macrophages and systemic infections. Maintains *Salmonella*-containing vacuole (SCV) as an intracellular niche for the survival and proliferation of the bacterium. Involved in phagosome tubulation and other alterations by translocating proteins.	[70,71]
*SPI-3*	Ensure survival in macrophages and growth in Mg-deficient environments. It encodes for the *MgtCB* (Magnesium transport system) operon, essential for bacterial survival in nutritionally deprived conditions. It also encodes for MisL protein (an anti-transport protein) that is essential for adhesion and long-term intestinal survival of *Salmonella*.	[71,72]
*SPI-4*	Possess genes for toxin secretion, apoptosis and intramacrophage survival. Responsible for gastrointestinal inflammation, required for adhesion to epithelial cells as well as membrane ruffle formation.	[73,74]
*SPI-5*	Possess genes that encode for T3SS effector proteins. Plays a role in enteropathogenicity and encodes for the proteins that are associated with intestinal mucosal fluid secretion as well as inflammatory response.	[75,76]
*SPI-6*	Responds to external stimuli to transport proteins to host cells. Possesses the *saf* gene (encodes for fimbriae) and the *pagN* gene (encodes for invasion protein). Essential for intramacrophage survival and successful establishment of the bacterium in the host’s gut during infection.	[76,77]

**Table 3 microorganisms-13-02363-t003:** Other virulence factors involved in *Salmonella* infection.

Virulence Factor	Location	Function	References
Fimbriae (adhesins)	Bacterial cell surface	Adhesion to host cell, biofilm formation, seroconversion, hemagglutination, cellular invasion and macrophage interactions	[78]
Flagella	Bacterial cell surface	Display flagellin phase variation, creating phenotypic heterogeneity of the flagellar antigens, which minimizes the host immune response towards the pathogen.	[79]
Surface polysaccharides	Bacterial cell surface	Allow persistence of the bacteria in the gut of the hosts.	[47]
Type III secretion system	Present on the bacterial cell surface. Encoded on *SPI-1*, *SPI-2* and other *SPIs*	Modification of host cell biology and successful infection by secreting several effector proteins into the host cell.	[39]
hylE protein (product of *hylE* gene)	Outer membrane of a bacterial cell	Pathogenesis of systemic salmonellosis is utilized in subserovar-level typing.	[80]
*Salmonella* plasmid virulence locus	Located on the *Salmonella* virulence plasmid.	Multiplication of *Salmonella* in the reticuloendothelial system.	[80]

**Table 4 microorganisms-13-02363-t004:** Probiotics’ effectiveness against *Salmonella* serotypes in broilers.

Probiotic Strain(s)	Administration	*Salmonella* Strain	Effect on *Salmonella* Colonization	References
*B. subtilis* QST-713	In feed	*S.* Gallinarum	Reduction in *Salmonella* content was not quantified.	[98]
*B. subtilis* DSM17299	In feed	*S.* Heidelberg	3 log_10_ reduction in cecal *Salmonella* content.	[99]
*B. subtilis*	In feed	*S.* Gallinarum	Reduction in *Salmonella* content was not quantified.	[101]
*B. subtilis* RX7 or *B. methylotrophicus* C14	In feed	*S.* Gallinarum	Reduction in *Salmonella* content was not quantified	[102]
*B. subtilis* B2A	In feed	*S.* Gallinarum	Reduction in *Salmonella* content was not quantified	[103]
Three-strain *Bacillus* probiotic	In feed	*S.* Enteritidis	1.08 log_10_ CFU/g reduction in cecal *Salmonella* content.	[106]
*B. licheniformis*, *B. subtilis*	In feed	*S.* Enteritidis	0.73, 1.59 and 1.32 log_10_ reduction at 5-, 12- and 21-days post-infection.	[15]
*B. coagulans*	In feed	*S.* Enteritidis	0.24, 0.41, and 0.24 log_10_ less *Salmonella* counts after 7-, 17-, and 31-days post-infection in treated birds than untreated birds.	[109]
*B. coagulans*	In feed	*S.* Enteritidis	0.40 and 0.60 log_10_ reduction in cecal *Salmonella* content, 0.80 and 0.75 log_10_ reduction in fecal *Salmonella* content	[113]
*B. subtilis KKU213*, *P. pentosaceus NP6*	In feed	*S.* Typhimurium	Complete elimination of *Salmonella* content was observed in treated birds (on day 18 post-treatment)	[115]
*P. freudenreichii* B3523	In drinking water	*S.* Heidelberg	1–2 log_10_ CFU/g reduction in cecal *Salmonella* content	[118]
*P. freudenreichii* B3523	In drinking water	*S.* Reading, *S.* Agona and *S*. Saintpaul	1.4–2.0 log_10_ CFU/g reduction in cecal *Salmonella* content	[119]
*Toyocerin* containing 10^10^ viable spores of *B. cereus var.* toyoi NCIMB 40112/CNCM I-10^12^ per gram	In feed	*S.* Enteritidis	100% reduction in cecal *Salmonella* content	[111]
*L. plantarum* and *B. subtilis*	In feed	*S.* Typhimurium	complete elimination of *Salmonella* in the liver, spleen and heart of the chickens 28 days post-infection	[121]
*Lavipan* (*Lactobacillus* spp. + yeast)	In feed	*S.* Enteritidis	Approx. 98.84% reduction in cecal *Salmonella* content compared to control (58.29%) on 42 days of life.	[122]
*L. plantarum*	In feed	*S.* Enteritidis	No reduction in *Salmonella* content	[123]
*L. acidophilus*, *L. casei*, *B. bifidum*, *Aspergillus oryzae*, *Streptococcus faecium* and *Torulopsis* spp.	In feed	*Salmonella* was isolated from carcasses.	40% carcass meat tested positive for *Salmonella*, as compared to 100% prevalence in carcass meat from untreated birds	[125]
*S. cerevisiae var. boulardii*	In feed	*S.* Enteritidis	Led to a 28.6% and 33.3% reduction in *Salmonella* content on breast skin and cloacae.	[129]
*B. amyloliquefaciens*, *B. licheniformis*, and *B. pumilus*, yeast culture, and yeast cell wall	In feed	*S.* Enteritidis	Led to a 0.79 log_10_ MPN/g reduction in the probiotic group as compared to the control, and a 0.86 log_10_ MPN/g reduction in the yeast culture-treated group as compared to the control.	[127]
*L. plantarum*, *L. acidophilus* and *S. cerevisiae*	In drinking water	*S.* Enteritidis	No reduction in cecal *Salmonella* content was observed	[128]

**Table 5 microorganisms-13-02363-t005:** Key virulence factors associated with *Campylobacter jejuni*.

Structure	Function	Initial Role	References
Flagella	Motility	The absence of either *FlaA* or *FlaB* leads to changes in the filament construction and the inability to move	[159]
	Binding and adhesion	Non-motile mutants exhibit less adherence	[143]
	Invasion	Lack of flagellins leads to the absence of infectivity in chicks	[160]
	Colonization	Lack of flagellar apparatus leads to insufficient colonization of the mouse intestines	[161]
	Secretory system	Aflagellated mutants fail to secrete invasion antigens as a Cia	[162]
	Immune evasion	Modified flagella evade recognition by toll-like receptor 5 (TLR5)	[163]
CheY	Chemotaxis	Absence of chemotaxis protein CheY leads to loss of invasiveness and motility	[164]
LOS	Immune evasion	Mimicry with nerve ganglioside initiates Guillain-Barré syndrome	[143]
		Evade antibody recognition and modulate immune response	[165]
CadF	Binding	Fibronectin-binding outer membrane protein	[162]
Cia	Invasion	Inhibition of Cia secretion significantly reduces the severity of *C. jejuni* in vivo	[166]
CDT	Toxins	CDT, particularly *CdtB*, causes apoptosis and cell cycle arrest	[167]

**Table 6 microorganisms-13-02363-t006:** In vitro screening of probiotics with inhibitory activities against *Campylobacter jejuni*.

Effectors	Test	Mechanism	Aim	References
Probiotics’ metabolites (cell-free supernatant)	- Well diffusion assay on agar - Cell line	Organic acids Bacteriocins	Testing their direct effect on *C. jejuni* or indirectly through modulating host immunity	[169]
Live culture of probiotics	- Agar slab test - Agar spot test - Co-culture suspension assay - Co-culture with the GIT cells monolayer - Adhesion and invasion assays	Auto-aggregation Co-aggregation Organic acids production Bacteriocins production	Testing probiotic candidates’ ability to: - Direct anti-*C. jejuni* activities. - Aggregation with *C. jejuni*, making a clump to reduce its invasion and adhesion - Competitive attachment to intestinal epithelial sites	[135,170,171,172]
Probiotics stability	- Acid tolerance - Bile tolerance	Resisting lower acids Resisting bile salts	Testing probiotic candidates’ ability to tolerate harsh gut conditions such as lower pH and bile salts.	[173,174]
Probiotics safety concern	Antibiotic sensitivity test	Lack of antibiotic resistance genes	Testing the absence of antibiotic-resistance genes	[175,176]

**Table 8 microorganisms-13-02363-t008:** Classification of *C. perfringens* toxinotypes relevant to poultry and humans.

Toxinotype	α-Toxin	β-Toxin	ε-Toxin	ι-Toxin	Enterotoxin	NetB	Associated Diseases	References
Genes	*cpa/plc*	*cpb*	*etx*	*Iap/ibp/itx*	*cpe*	*netB*		
A	+	-	-	-	-	-	Gas gangrene, food poisoning, NE in fowls	[209]
C	+	+	-	-	+/-	-	Necrotizing enteritis in humans	[208]
F	+	-	-	-	+	-	Food poisoning and abdominal cramps	[208]
G	+	-	-	-	-	+	Avian NE	[208]

**Table 10 microorganisms-13-02363-t010:** Virulence factors of different pathogenic *Escherichia coli* strains.

Virulence Factor	Role in Pathogenesis	References
Bundle-Forming Pili (BFP)	Helps bacteria adhere to each other to form a colony and facilitates the initial interaction between bacteria and host epithelial cells. Allows for the bacteria to attach in a localized manner.	[235]
Longus Pillus	Promotes intestinal colonization by aiding bacterial adherence and enabling mobility through twitching motility.	[236]
Curli fimbriae	Formation of biofilms on organic and non-organic surfaces in *E. coli* O157:H7. Facilitate the attachment of *E. coli* to host epithelial and endothelial cells, thereby enhancing bacterial dissemination and persistence in host tissues and the bloodstream.	[237]
Aggregative Adhesion Fimbriae (AAF)	Enteroaggregative *E. coli* exhibit agglutination and adhere to components of the extracellular matrix	[238]
Translocated intimin receptor (Tir) protein	Binds to intimin on the host cells to enable attachment and effacing lesions characteristic of enterohaemorrhagic and enteropathogenic strains of *E. coli*	[239]
Injectisome protein	Connects the pathogenic bacteria to the eukaryotic host cells, and allows proteins to enter enterocytes	[239]
STa Oligopeptide Toxin and A1B5 Heat-Labile Enterotoxin	Induces disruption of electrolyte and water homeostasis by modulating ion transport mechanisms, leading to watery diarrhea	[240]
Vero toxin or Shiga toxin	Interferes with protein synthesis by targeting ribosomal function to cause blood vessel and tissue damage within the intestinal and renal systems	[241]
HlyA toxin	Contributes to tissue invasion by lysing red blood cells and host cells, promoting bacterial dissemination, thus damaging host tissues.	[242]

**Table 11 microorganisms-13-02363-t011:** Probiotics’ effectiveness against *E. coli* serotypes in broiler chickens.

Type of Probiotic	Administration	Probiotic Dose	*E. coli* Challenge Strain	Effect on *E. coli* Colonization	References
*L. plantarum* B1	In feed	2 × 10^12^ CFUs/kg added to basal diet	*E. coli* K88	Decreased *E. coli* colonization in cecal digesta by 0.3 log_10_	[260]
*L. bulgaricus*	Oral gavage	3 × 10^9^ CFU/mL	*E. coli* O157:H7	Decreased colonization in the cecum	[261]
*Bacillus subtilis* (Calsporin^®^)	In feed	0.1% *B. subtilis* in diet	*E. coli* strain PTCC-1399	Reduction in *E. coli* colonization by 1.7 log_10_ CFUs/g	[262]
*B. subtilis* PY79	In feed	2 × 10^9^ CFU/kg of diet	*E. coli* O78:K80	Decreased *E. coli* colonization by 3.2 × 10^7^ CFUs/g	[248]
*E. faecalis*-1	Oral gavage daily for three days	1 × 10^8^ CFU	*E. coli* O78	No significant change	[225]
Microencapsulated *E. faecalis*	In feed	1 × 10^10^ CFU/kg of diet	*E. coli* K88	Reduction in *E. coli* colonization by 0.5 log_10_ CFUs/g	[250]
*E. faecium* NCIMB11181	In feed	5.1 × 10^10^ CFU/kg of diet	*E. coli* O78	*E. coli* reduction of approximately 2 log_10_ CFUs/g in liver	[251]
*E. faecium* HJEF005	In feed	10^9^ CFUs/kg of feed	*E. coli* K88	*E. coli* reduction of approximately 0.27 log_10_ CFUs/g in the cecum	[252]
*C. butyricum* HJCB998	In feed	2 × 10^7^ CFUs/kg of feed	*E. coli* K88	Decreased cecal *E. coli* colonization by approximately 1 log_10_ CFUs/g	[263]
Probac Plus *(L. acidophilus*, *L. plantarum*, *L. bervis*, *L. bifidobacterial*, and *S. cerevisae)*	In feed for 1–7 days	0.5 g/kg	*E. coli* K88	Decreased colonization by approximately 5.4 × 10^6^ CFUs	[264]
*S. cerevisiae*	In feed	Not specified	*E. coli* strain isolated from infected birds	Reduced *E. coli* fecal load by 2.4 × 10^8^ CFUs	[265]
*Lacticaseibacillus rhamnosus* GG	Oral inoculation	10^8^ CFUs/bird	*E. coli* O78	Reduction of *E. coli* colonization by approximately 1.6 log_10_ CFUs in the cecum, and inhibition of colonization in the heart and liver	[257]
*Lacticaseibacillus rhamnosus* GG ATCC 53103	In feed	10^6^ CFUs/g of feed	*E. coli* isolated from clinically infected ducks	Reduction of *E. coli* colonization by approximately 1.5 log_10_ CFUs/g in the heart, 1 log_10_ CFUs/g in the lungs, and 0.5 log_10_ CFUs/g in the kidneys	[256]
*B. lactis* Bb12	Oral inoculation	10^8^ CFUs/bird	*E. coli* O78	Reduction of *E. coli* colonization by approximately 0.6 log_10_ in cecal contents	[257]
*Mixed strain (Bacteroides caecicola*, *Bacteroides plebeius*, *Megasphaera stantonii*, *Megamonas hypermegale*, *Megamonas funiformis*, *Phascolarctobacterium faecium*, and *Sutterella massiliensis)*	Oral inoculation	0.1 mL of gut anaerobes	*E. coli* O78:H4 ST117	Inhibition of pathogenic *E. coli* growth in experimental chickens (with one exception)	[266]
*L. acidophilus*, *B. subtilis*, *Bacillus megaterium*, *L. bulgaricus*, *Candida pintolopesii*, and *S. cerevisiae*, *Aspergillus oryzae*, and *Streptococcus thermophilus*	Drinking water	10^12^ CFUs/g	APEC	Noted increase in mortality, potentially due to disease and decreased immunity	[267]

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
