# Peer review of "Major Foodborne Bacterial Pathogens in Poultry: Implications for Human Health and the Poultry Industry and Probiotic Mitigation Strategies"

_microorganisms, 2025, doi:10.3390/microorganisms13102363_

Round 1
Reviewer 1 Report
Comments and Suggestions for Authors
Major Foodborne Pathogens in Poultry: Implications for Human Health and the Poultry Industry and Probiotic Mitigation Strategies.
L37: update statistics to 2025.
L38: not accurate information. In 2024, the US produced 9.33 billion broiler chickens, resulting in 61.1 billion pounds of live weight.
You need to provide a paragraph regarding table eggs and turkey in the introduction, since you discuss eggs in some parts of the review.
Revise the introduction with more accurate information, updated statistics.
L48-49: Provide more specific information, the year.
L50-52: Is it all from poultry?
L71-73: provide the year
L86-87: not accurate information
L83-85: You need to distinguish between therapeutic and subtherapeutic antibiotics.
L144-145 & L141: Invasive non-typhoidal Salmonella (iNTS), here and everywhere, you need to define all abbreviations once they appear in the paper.
L152-159 & L160-181: Move to the beginning of the Salmonella heading
The paragraphs in general need to be more organized. Give information about the bacteria, then the disease.
Table 1: add more references; all the information was taken from one source. Also, in Table 2, you need to add more references. You need to collect the material for the review from different sources and create the table based on all the gathered information.
L235-368: You need to provide some justification and explain the mechanism of action for those probiotics you list. Provide other studies that showed no effect as well. Include studies with layers as well.
The previous note applies to all bacteria you have used in your review.
L576: add subheadings for other bacteria, C. perfringens
Table 8: just provide studies for poultry.
Table 10. Virulence factors of different pathogenic E. coli strains. add more references; all the information was taken from one source.
There were so many important studies on this topic that were not cited in this review.
Author Response
We sincerely thank the reviewer for their thoughtful and constructive feedback. We have carefully considered each of the comments and provided a detailed, point-by-point response and all the changes made to the manuscript are highlighted in yellow
The reviewer’s insights have been invaluable in refining the manuscript, clarifying our arguments, and strengthening the overall quality and impact of the article. L37: update statistics to 2025.
The statistics have been updated
L38: not accurate information. In 2024, the US produced 9.33 billion broiler chickens, resulting in 61.1 billion pounds of live weight.
Thank you for the thorough review. We have corrected the inaccurate information as per your suggestion.
You need to provide a paragraph regarding table eggs and turkey in the introduction, since you discuss eggs in some parts of the review.
Information for table eggs and turkey has been added in the introduction as suggested.
Revise the introduction with more accurate information, updated statistics.
The introduction has been thoroughly revised, and the statistics have been updated
L48-49: Provide more specific information, the year.
Specific information was added.
L50-52: Is it all from poultry?
We have verified the cited information and made the necessary changes
L71-73: provide the year
We have added the year as suggested
L86-87: not accurate information
The cited information has been verified, and the necessary amendments have been made accordingly
L83-85: You need to distinguish between therapeutic and subtherapeutic antibiotics.
We were referring to subtherapeutic use. The statement has been revised for clarity. Thank you for the thorough review.
L144-145 & L141: Invasive non-typhoidal Salmonella (iNTS), here and everywhere, you need to define all abbreviations once they appear in the paper.
The manuscript has been thoroughly revised, and all abbreviations have been checked to ensure each appears only once.
L152-159 & L160-181: Move to the beginning of the Salmonella heading
The paragraphs in general need to be more organized. Give information about the bacteria, then the disease.
We fully agree with the reviewer. The paragraphs describing Salmonella and its serotypes have been moved to the beginning of the Salmonella section.
Table 1: add more references; all the information was taken from one source. Also, in Table 2, you need to add more references. You need to collect the material for the review from different sources and create the table based on all the gathered information.
We thank the reviewer for their valuable input. The tables have been updated, and the revised manuscript now includes additional sources.
L235-368: You need to provide some justification and explain the mechanism of action for those probiotics you list. Provide other studies that showed no effect as well. Include studies with layers as well.
While the specific mechanisms of action of each probiotic bacterium remain unknown, we have provided a general mechanism of action for probiotics in the Introduction and have discussed their effects on each pathogen within the respective sections. We acknowledge that this presentation may appear biased toward positive results; however, we want to emphasize that we did not interpret these effects as strictly positive or negative. Rather, we reported the observed log reductions as presented in the original studies, which varied widely, ranging from 0.19 to 3 log reductions.
The previous note applies to all bacteria you have used in your review.
L576: add subheadings for other bacteria, C. perfringens
As suggested, we added a subheading for each bacterium covered in this review article, including C. perfringens. Your suggestion has greatly improved the organization of our manuscript. Thank you.
Table 8: just provide studies for poultry.
We have removed the table, as the data related to poultry is already discussed in the text. The table was originally included to illustrate differences in C. perfringens toxins among different species.
Table 10. Virulence factors of different pathogenic E. coli strains. add more references; all the information was taken from one source.
Thank you. The table has been updated and data from different sources have been added as suggested.
There were so many important studies on this topic that were not cited in this review.
We have added additional studies to the tables to include as many relevant references as possible. While we agree that many other studies could be included, it is challenging to incorporate all of them, as the article is already extensive. Thank you for your valuable input.
Reviewer 2 Report
Comments and Suggestions for Authors
The manuscript entitled "Major Foodborne Pathogens in Poultry: Implications for Human Health and the Poultry Industry and Probiotic Mitigation Strategies" is an interesting and well-organized study.
Comments and Suggestions for Authors
Poultry production is rapidly growing, but it increases the risk of foodborne pathogens like Salmonella, Campylobacter, Escherichia coli, Clostridium perfringens, and Listeria. Antibiotic growth promoters (AGPs) have been used to mitigate disease, but concerns over antimicrobial resistance have led to interest in probiotics as a preharvest intervention. This review investigates major foodborne pathogens associated with poultry and evaluates the practical implementation of probiotic-based strategies in modern poultry production systems, with the goal of reducing pathogen load and enhancing overall food safety.
Abstract:
- The abstract describes the sections of the manuscript well.
Introduction:
- These data effectively highlight the dual nature of poultry's success: its crucial role in global nutrition is complemented by its substantial contribution to the spread of foodborne illnesses. This underscores the need for effective alternatives and the economic burden on public health systems and the poultry industry. The transition from pathogens and costs to historical solutions and drawbacks outlines the need for sustainable production interventions.
- The introduction highlights the multifaceted effects of probiotics, including inhibitory compounds, immune modulation, and gut integrity improvement. It emphasizes rigorous selection criteria for true probiotics and innovative delivery methods, like in ovo administration, providing a primer on their scientific rationale in poultry production.
- The introduction, although well-written, is criticized for being too lengthy, so it should be shortened.
- The sections of the manuscript addressed the objectives of the review well, so I recommend publishing it in Microorganisms.
What is the main question addressed by the research?
The authors did adequately explain the purpose of the study.
Do you consider the topic original or relevant to the field? Does it address a specific gap in the field? Please also explain why this is/ is not the case.
- The topic is highly relevant to the fields of poultry science, animal nutrition, and food safety. It is moderately original, not for introducing the concept of probiotics, but for its proposed methodology to solve a critical, persistent problem in the field. Additionally, it directly addresses two of the biggest challenges in modern poultry production: preharvest food safety and antimicrobial reduction.
- The originality lies not in the use of probiotics itself, but in the proposed solution. The shift from using a few traditionally cultured probiotic strains to developing a complex, well-defined, poultry-specific consortium based on advanced multi-omics data is a novel and cutting-edge approach. It moves the field from a trial-and-error method to a rational, hypothesis-driven design.
What does it add to the subject area compared with other published material?
I think the authors covered the subject adequately. They provided a comprehensive analysis that highlights key points while addressing potential counterarguments. Overall, their insights contribute significantly to the ongoing discussion in the field.
What specific improvements should the authors consider regarding the methodology?
No comments.
Are the conclusions consistent with the evidence and arguments presented and do they address the main question posed? Please also explain why this is/is not the case.
Yes.
Are the references appropriate?
It's enough.
Any additional comments on the tables and figures?
No comments.
Author Response
Comments and Suggestions for Authors
Poultry production is rapidly growing, but it increases the risk of foodborne pathogens like Salmonella, Campylobacter, Escherichia coli, Clostridium perfringens, and Listeria. Antibiotic growth promoters (AGPs) have been used to mitigate disease, but concerns over antimicrobial resistance have led to interest in probiotics as a preharvest intervention. This review investigates major foodborne pathogens associated with poultry and evaluates the practical implementation of probiotic-based strategies in modern poultry production systems, with the goal of reducing pathogen load and enhancing overall food safety.
Abstract:
- The abstract describes the sections of the manuscript well.
Thank you for your positive feedback.
Introduction:
- These data effectively highlight the dual nature of poultry's success: its crucial role in global nutrition is complemented by its substantial contribution to the spread of foodborne illnesses. This underscores the need for effective alternatives and the economic burden on public health systems and the poultry industry. The transition from pathogens and costs to historical solutions and drawbacks outlines the need for sustainable production interventions.
- The introduction highlights the multifaceted effects of probiotics, including inhibitory compounds, immune modulation, and gut integrity improvement. It emphasizes rigorous selection criteria for true probiotics and innovative delivery methods, like in ovo administration, providing a primer on their scientific rationale in poultry production.
- The introduction, although well-written, is criticized for being too lengthy, so it should be shortened.
We thank the reviewer for their feedback and have shortened the introduction as per your valuable suggestion.
- The sections of the manuscript addressed the objectives of the review well, so I recommend publishing it in Microorganisms.
We appreciate your valuable feedback and your endorsement of our article for publication
What is the main question addressed by the research?
The authors did adequately explain the purpose of the study.
Thank you
Do you consider the topic original or relevant to the field? Does it address a specific gap in the field? Please also explain why this is/ is not the case.
- The topic is highly relevant to the fields of poultry science, animal nutrition, and food safety. It is moderately original, not for introducing the concept of probiotics, but for its proposed methodology to solve a critical, persistent problem in the field. Additionally, it directly addresses two of the biggest challenges in modern poultry production: preharvest food safety and antimicrobial reduction.
- The originality lies not in the use of probiotics itself, but in the proposed solution. The shift from using a few traditionally cultured probiotic strains to developing a complex, well-defined, poultry-specific consortium based on advanced multi-omics data is a novel and cutting-edge approach. It moves the field from a trial-and-error method to a rational, hypothesis-driven design.
We concur with the reviewers’ view on this research concept.
What does it add to the subject area compared with other published material?
I think the authors covered the subject adequately. They provided a comprehensive analysis that highlights key points while addressing potential counterarguments. Overall, their insights contribute significantly to the ongoing discussion in the field.
We truly appreciate your positive feedback.
What specific improvements should the authors consider regarding the methodology?
No comments.
Are the conclusions consistent with the evidence and arguments presented and do they address the main question posed? Please also explain why this is/is not the case.
Yes.
Thank you.
Are the references appropriate?
It's enough.
Thank you.
Reviewer 3 Report
Comments and Suggestions for Authors
This review systematically introduces the major foodborne pathogens related to poultry and assesses the actual implementation of probiotics in modern poultry production systems. Overall, the logic is clear and the information is abundant, which has certain reference value in the field of poultry production. However, the structure of the article can be further optimized. I propose the following suggestions.
- The introduction part is okay, but it is a bit lengthy. It can be made more concise.
- In the second part, each chapter is too long, making it a bit difficult to read. It is suggested that the author add some third-level heading under each second-level heading. Each pathogens can be sorted according to its classification, transmission route, pathogenic mechanism, types of probiotics targeted against the pathogen, the effects and mechanisms of the probiotics, etc.
Author Response
This review systematically introduces the major foodborne pathogens related to poultry and assesses the actual implementation of probiotics in modern poultry production systems. Overall, the logic is clear and the information is abundant, which has certain reference value in the field of poultry production. However, the structure of the article can be further optimized. I propose the following suggestions.
The introduction part is okay, but it is a bit lengthy. It can be made more concise.
We thank the reviewer for their feedback and have shortened the Introduction as per your valuable suggestion.
In the second part, each chapter is too long, making it a bit difficult to read. It is suggested that the author add some third-level heading under each second-level heading. Each pathogens can be sorted according to its classification, transmission route, pathogenic mechanism, types of probiotics targeted against the pathogen, the effects and mechanisms of the probiotics, etc.
As suggested, we have added subheadings for each bacterium covered in this review article. Your suggestion has greatly improved the organization of our manuscript.
Round 2
Reviewer 1 Report
Comments and Suggestions for Authors
No further comments
Author Response
We sincerely thank the reviwer for their positive and constructive feedback.